# Insurance engagement in flood risk reduction – examples from household and business insurance in developed countries

Isabel Seifert-Dähnn

Norwegian Institute for Water Research, Gaustadalléen 21, 0349 Oslo, Norway

*Correspondence to*: Isabel Seifert-Dähnn (Isabel.Seifert@niva.no)

**Abstract.** Insurance can be an important mechanism to stimulate flood risk reduction and thus decrease losses. However, there is a gap between the theoretical potential described by academic scholars and the actual engagement of insurers. In the analysis, I have collected examples of insurers engagement in flood risk reduction, focusing on household and business insurance in developed countries. Insurers engaged either directly e.g. through co-financing risk reduction or more indirectly by giving

incentives to policyholders or governmental actors to adopt risk reduction measures. I analyzed their engagement with the framing conditions of the market they were acting in such as market penetration or private or public insurance scheme. I found risk reduction measures like awareness-raising campaigns targeting citizens to be quite common across several countries. There was less insurance engagement in risk reduction measures such as warning or land-use planning, which are perceived to be mainly governmental tasks. The use of risk-based pricing as an incentive for the adoption of risk reduction measures as

suggested by academia is difficult in practice, due to barriers such information gaps on the effectiveness of property-level protection measures and requirements concerning the affordability of insurance. New approaches to overcome these shortfalls include organized data collection on property-level protection measures or the insurance of high-risks for affordable premiums in public-private partnership constellations with the government.

## 1    Introduction

Economic losses of weather-related hazards are already high and expected to increase in the future (CEA, 2009; Jongman et al., 2014; Paudel et al., 2015; Swiss Re, 2012). Between 2000 and 2016, hydrological events, i.e. flooding and mass movements, caused 123 billion USD of overall losses, while only 36 billion USD were insured (Munich Re, 2017). Mainly socio-economic developments, but also climate change can largely be held accountable for rising loss trends, with valuable

assets increasingly exposed to flood risks (Alfieri et al., 2016; Botzen et al., 2010; Hoeppe, 3; Kundzewicz et al., 2014; Morita, 2014). The negative impacts of rising flood losses challenge governments, the public and the private sector to develop sustainable flood risk management mechanisms aimed at reducing those losses (Michel-Kerjan and Kunreuther, 2011; Mills, 2005). Acknowledging the slow progress and limited success of climate change mitigation in reducing greenhouse gases, sustainable adaptation is seen as one necessary management strategy to reduce and manage the risks of climate change

(Eisenack et al., 2014; IPCC, 2014). Adaptation is defined as adjustment to actual or future implications of climate change aiming to avoid harm or to exploit benefits (IPCC, 2014). It should be noted that in natural hazard research the terms 'adaptation' and 'mitigation' are often used synonymously in the sense of reducing impacts or losses (see e.g. Bouwer et al. (2014)). In this paper, only the term 'adaptation' is used.

5    Flood risk is generally determined by the hazard itself, the population, assets and values at risk from being flooded (exposure), and the vulnerability of society, i.e. their capacity or ability to cope with flood events (Kron, 2005). Thus, flood risk reduction measures can be classified according to which of the flood risk determinants they influence (Figure 1, right). Another important aspect is that different flood reduction measures address floods of different strength and return periods (Figure 1, left): while emergency management and adaptation measures are best suited to avoid or reduce losses from minor flood events with a high

10   return period, successful land-use planning has the potential to avoid not only losses from minor events, but also from more seldom extreme events. Risk knowledge and warning are considered pre-conditions, which facilitate or enable the implementation of the other risk reduction measures. Insurance has a special role in this context, as it is not primarily meant to avoid or reduce losses, but to compensate for losses after seldom severe events and thus facilitate recovery. Flood risk management today relies on a mix of structural and non-structural measures (Fuchs et al., 2017; Kreibich et al., 2015;

15   Krysanova et al., 2008; Kubal et al., 2009).

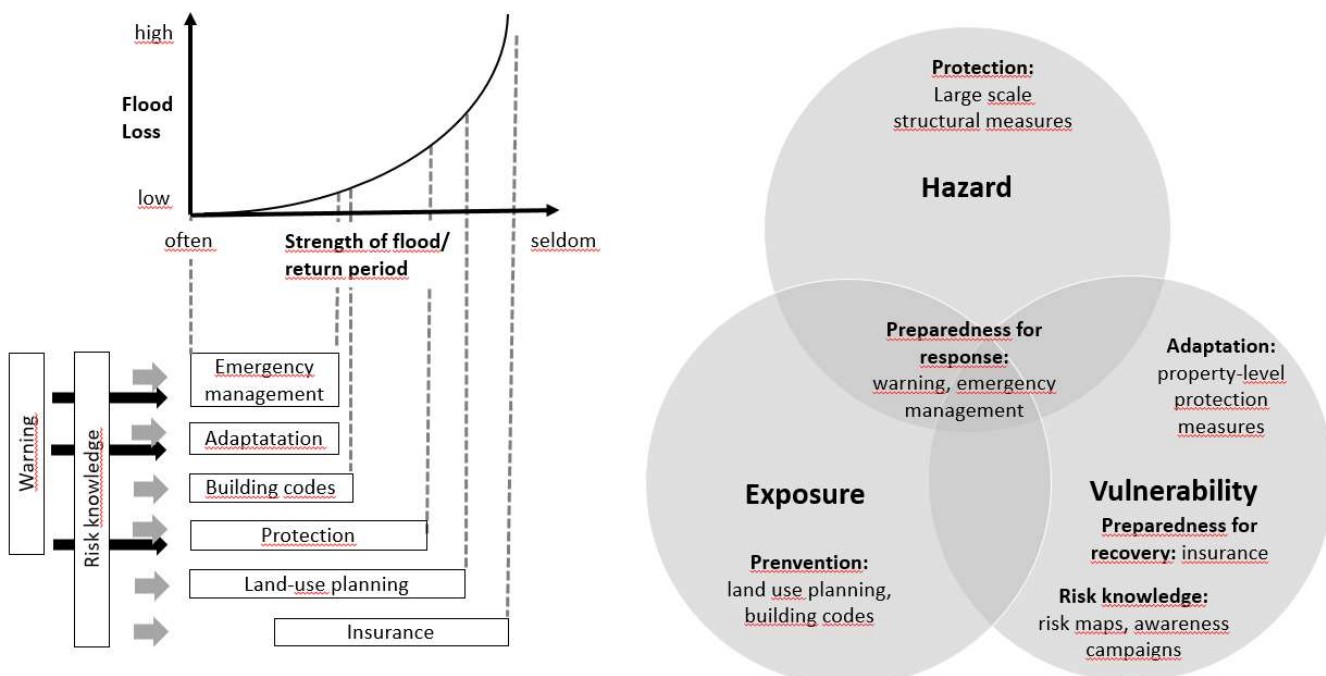

**Figure 1 Flood risk reduction measures sorted by their potential to reduce losses due to the strength of the flood event (left) and sorted after their risk reduction mechanism (right)**

Flood risk management is often framed as a cyclical process (see figure 2, lower right circle) starting with a flood event (Thieken et al., 2016). As direct response to the flood event i.e. already during the flooding emergency measures can be taken to limit negative impacts as e.g. the destruction of valuable items. In the recovery stage i.e. when the flood is over, damages are repaired, and affected societies will try to come back to "normal life" or pre-flood conditions. In the risk assessment phase, the flood risk is assessed in a systematic way to inform planning of risk reduction measures in the risk reduction phase. In reality, these phases are often linked with each other i.e. risk reduction measures can be implemented while damages are repaired (Moatty, 2017).

Actors involved in flood risk reduction span from state and local government to municipalities, private households and businesses and insurers. Governmental actors decide on a country's insurance scheme, regulate the market, set and enforce building standards allowing properties to withstand flooding, develop land-use plans, implement large-scale flood protection measures and provide warning and emergency services. Households and private business can contribute to flood risk reduction by preventing losses from minor high-frequent flood events on their own property and by purchasing insurance to cover larger losses. The primary task of insurance is risk transfer, i.e. the spreading of losses in time and space, shared with third parties in exchange for a premium (Bouwer et al., 2014; Duus-Otterström and Jagers, 2011; Warner et al., 2009). If working as supposed to, insurance gives affected people fast access to capital for reconstruction, enabling fast recovery in the aftermath of a disaster. Without insurance many activities would simply be too risky to be undertaken (Ranger et al., 2011).

Beyond this core function of insurance, there is growing recognition of the large potential that insurance has in providing incentives for other risk reduction measures and to prevent damages (Botzen et al., 2010; Box et al., 2016; Bräuninger et al., 2011; Herweijer et al., 2009; Kunreuther, 1996; Surminski and Eldridge, 2015). This aspect also received attention among policy makers: The European Union asks in its Green Paper on insurance of natural and man-made disasters (EC, 2013), how risk transfer mechanisms can better fulfill their prevention role (Surminski, 2014). The re-insurance industry also appeals to combine flood insurance and prevention (Swiss Re, 2012). Instead of being common practice, the link between insurance and flood risk reduction, to date, appears to be more theoretical in nature, i.e. insurance engagement in flood risk reduction seems to be quite limited (Den et al., 2017; Smolka, 2006; Surminski and Hudson, 2017). In addition, insurance can also lead to a moral hazard i.e. having a detrimental effect by creating disincentives for flood risk reduction (O'Hare et al., 2015).

Flood insurance schemes in developed countries vary considerably (see Table 1) with respect to who provides insurance, the degree of government involvement and cooperation with insurers, legal requirements, the availability and demand for insurance, if insurance is compulsory or voluntary, in their design of insurance products, in concerning their market penetration (Atreya et al., 2015; Bouwer et al., 2007; Den et al., 2017; Johannsdottir, 2017; Lamond and Penning-Rowsell, 2014; Porrini and Schwarze, 2014; Surminski et al., 2015; Suykens et al., 2016). These framing conditions for insurance businesses are due not only to a country's overall flood risk (Feyen et al., 2012), but also historic developments of insurance schemes and national societal preferences of how disaster losses should be shared. As there is no 'one size fits all' approach for flood insurance, it can be expected that the different framing conditions influence insurers engagement in flood risk reduction and the incentives they use to foster the uptake of flood risk reduction measures by other actors i.e. governments, private persons and business.

**Table 1. Overview of insurance schemes and flood insurance conditions for household and business insurance in several countries depicting the range from public to private insurance schemes;** Information in this table was derived from Atreya et al. (2015); Den et al. (2017); Guillier (2017); Hanger et al. (2017); Kousky (2017); Kousky et al. (2016); Priest et al. (2016); Surminski and Eldridge (2015); Surminski and Thieken (2017)

| Country | Insurance Scheme public-private | Bundled or single hazard | Flat or risk-based premium | Compulsory/quasi-compulsory or voluntary | Market penetration |
|---|---|---|---|---|---|
| Switzerland | Public monopoly insurer in 19 cantons, private insurance in 7 cantons | Bundled multiple hazards | Flat premiums | Quasi-compulsory (tied with fire-insurance) | High (99% of all buildings) |
| Spain | Public, but customer relationship managed by private insurance companies | Bundled multiple hazard | Flat premiums | Compulsory | 75-100% (for households and business) |
| US | Public, but customer relationship managed by private insurance companies | Single hazard | Risk-based premiums are applied for 80% of all policies | Quasi-compulsory in the 100-year floodplain (requirement for mortgages from regulated or state backed lenders), otherwise voluntary | Large variations between the states, maximum 35% in Florida |
| France | Public-private; private primary insurers, public & private reinsurance | Bundled multiple hazards | Flat premiums of 12% surcharge on property insur. | Quasi-compulsory (tied with property insurance) | High (100% for households, 90% for business) |
| Denmark | Public (flooding from rivers and seas due to 1/20 years storms); Private for minor events | Bundled | Limited risk differentiation | Public: quasi-compulsory (tied with fire-insurance) Private: voluntary | 50-75% for households |

| Country | | | | | |
|---|---|---|---|---|---|
| Austria | Private insurance with limited coverage co-existing with a governmental post-disaster catastrophe fund | Bundled | No risk-based premiums | Voluntary | Low (10-25%), Often insurance is denied in high-risk areas |
| Sweden | Private | Bundled with property insurance | Limited risk differentiation | Quasi-compulsory (requirement for mortgages) | High (>75% for business, >95% for households) |
| UK (post 2016) | Private | Bundled with building and content insurance | Risk-based pricing, however little variation in premiums | Quasi-compulsory (requirement for mortgages) | High (75% -98% for households) |
| Germany | Private | Single hazard and bundled insurance available | Risk-based | Voluntary | Low (40% for building insurance) |

There is a small but steadily growing body of literature that examines certain aspects of this issue. Present studies have focused on which economic instruments can be used to incentivize prevention (Bräuninger et al., 2011; Filatova, 2014), how the insurance system should be designed to support more adaptation (Kunreuther, 1996; Lamond and Penning-Rowsell, 2014; Michel-Kerjan and Kunreuther, 2011), how prevention is or could be considered in the recovery process (Priest et al., 2016; Suykens et al., 2016), the distribution of responsibilities between insurers and the government (Keskitalo et al., 2014), how policy and market factors hamper or support insurers engagement in flood risk adaptation (Glaas et al., 2016; Surminski et al., 2015), and recently also how in certain countries insurers are engaged in flood risk reduction (Den et al., 2017; Poussin et al., 2013; Surminski, 2014; Surminski and Hudson, 2017; Surminski and Thieken, 2017a). Nevertheless, it remains still unclear to what extent insurance can be used to boost flood risk reduction (Surminski, 2014), for what types of flood risk reduction measures and what are the incentivizing mechanisms insurers currently use or could use (Atreya et al., 2015).

This article tries to shed a light on this, by investigating current engagement of insurers in developed countries in different flood risk reduction measures and their use of incentives to get other actors engaged. I analyze how these activities are influenced by framing conditions such as the insurance scheme or market penetration (see assessment framework depicted in figure 2). The study focuses on developed countries and on household and business flood insurance. Detailed legal requirements for insurers in each country were not considered as this was beyond the scope of this study. The analysis is not limited to selected countries, but there is certainly a bias in the number of publications about insurance systems in different countries and the access to national publications was limited by language skills of the author.

## 2    Context - How does flood insurance work?

Insurance is based on the principle of solidarity, i.e. individual persons get together to form a risk-bearing community in the case of losses, to cover the loss of its individual members. This function is currently organized and coordinated by insurance companies, which offer insurance to individuals in exchange for a premium. The collected premiums are then used to compensate individual losses. Insurers are organized in different legal forms, some are non-profit state-owned companies, other are private entities owned by their policyholders (cooperative/mutual insurance), while other are private profit-oriented companies.

In principle, risks are considered insurable when the following conditions are fulfilled: there is a **large enough number of exposed people** that perceive they might suffer a **considerable loss** that they cannot cope with alone. Subsequently, if they wish to share this risk with a third party, there is a demand for insurance. The loss itself must **happen randomly** and must **be evident**, i.e. it must be possible to demarcate it i.e. it happens at a certain time interval, at a defined place, and from a known cause. In order to offer insurance, insurers must be able to **calculate expected losses** by knowing their probability of occurrence as well as the costs attached to recovery. Historic claim data in combination with risk modelling is often used to assess this. Many risks occur with different frequencies and intensities and therefore insurance companies must be sure that there is **only limited risk that catastrophically large losses will occur**. Furthermore, insurers must be able to offer a product at an

**affordable premium** so that demand can be met. At the same time, **economic feasibility** must be known, i.e. the sum of premiums should at least be high enough to cover the expenses of the insurer (loss compensation plus administrative costs) or in case of private insurance companies, to also generate a profit.

These criteria must not be seen as absolute: catastrophically large losses are at least partly transferrable to reinsurance companies and financial markets, and sometimes governments take the role as insurer of the last resort as for example the Consorcio de Compensación de Seguros in Spain. Demand can be sufficiently created by bundling one type of risk with another (e.g. fire insurance is bundled with flood insurance in Belgium) or by making insurance compulsory for everyone (e.g. in France). Affordability for low-income groups can be accomplished by state subsidies; for example in the US, low-income households can receive premium discounts, although this practice has now been phased out (Kousky and Kunreuther, 2013).

Insuring against floods is a challenging issue (Swiss Re, 2012). First, it is technically difficult to assess exposure, probability of occurrence and potential losses. Climate change complicates this further as it is still largely unknown how climate change will impact flood risk in detail (Kundzewicz et al., 2014). Second, the risk-bearing community is often small: only people who perceive to be at risk from flooding, demand insurance. Another problem is **adverse selection** (Swiss Re, 2012); there is an information asymmetry between the policyholders and insurers; resulting in more people at high risk from flooding seeking insurance, while insurers charge a too low premium for this risk. To build up financial resources enough to cover potential damages would require high premiums, which in turn counteract the affordability criterion.

It is not surprising from a mathematical point of view that certain areas or properties are considered uninsurable. This is often where the government steps in with different forms of interventions. The degree to government intervention ranges from taxpayer financed flood loss compensation in the Netherlands, monopoly insurance systems as in parts of Switzerland[1] and Spain, compulsory all-natural-hazard or bundled insurances (e.g. France), to single-hazard mixed government-private sector systems (US) and private insurance markets with only limited or ad-hoc compensation by the government (e.g. Germany, UK). Flood insurance can also be quasi-compulsory, meaning that the insurance itself is voluntary, but for example, needed for taking up a property mortgage (Smolka, 2006). To explore this, several scholars have carried out comparative reviews of insurance systems in different countries (Atreya et al., 2015; Bräuninger et al., 2011; Den et al., 2017; Keskitalo et al., 2014; Lamond and Penning-Rowsell, 2014; Paudel, 2012; Porrini and Schwarze, 2014; Suykens et al., 2016).

Government interventions can have advantages and disadvantages. When losses are compensated by the tax-payer or by ad-hoc post disaster government assistance (as in Germany after severe flood events in 2002 and 2013) (Thieken et al., 2016) or people feel safe behind large structural protection measures this can lead to a problem called **charity hazard**: people at risk see less necessity to purchase insurance or undertake prevention measures as they presume their losses are already covered(Hudson et al., 2017). Similarly **moral hazard** will occur when obtained insurance coverage decreases peoples

---

[1] For Switzerland, it should be noted that most articles characterize the Swiss system as a monopoly insurance system (see e.g. (Bräuninger et al., 2011; Porrini and Schwarze, 2014). In fact, this is only true for 19 of the 26 cantons. In the remaining cantons, where natural hazard insurance is not provided by Cantonal Insurance Monopolies, private companies compete in a free-market system (Paudel, 2012).

motivation to take risk reduction measures (Hudson et al., 2017). Moral hazard can also occur at the government level, where high insurance market penetration or the availability of emergency funds (as from the European Solidarity Fund) might lower the urgency for the government to implement prevention measures (Surminski, 2014; Surminski et al., 2015).

Differences also exist in the design of flood insurance products. Coverage exists for damages to buildings, their contents, and cars, as well as to cover business interruption, loss of agricultural harvest or damages to infrastructure. While the household and small business sector can be considered as mass markets for flood insurance, for high-value objects such as large companies contract conditions are often negotiated separately. A further distinction is made between different types of flood, which is necessary to demarcate the event. Examples include riverine flood, coastal flood, storm surge, flash flood, torrential rainfall, dam burst, ice jam, mudflow, lahar, groundwater flooding or tsunami (Swiss Re, 2012). There is insurance that covers only one type of flood, several, or that bundle the flood risk together with other natural hazards, such as the French all-hazards insurance (Bräuninger et al., 2011). The insurance contract specifies the length of the contract, premium and deductibles, indemnity limits, coverage and exclusions (e.g. of certain types of events or certain assets) as well as special conditions (Surminski and Eldridge, 2015). These special conditions can require the implementation of preventive measures. In combination with the premium, deductibles and coverage adjustments are considered important incentives to trigger more preventive behavior (Kunreuther, 1996). The disadvantage of having many contracts with different contract conditions is, that the insurer is required to follow up thatthe conditions are fulfilled, which raises their **transaction costs** (Botzen and Van Den Bergh, 2008; Thieken et al., 2006). Nowadays one-year insurance contracts are most common (Bräuninger et al., 2011). This is often seen as a barrier for risk prevention, as most risk reduction measures only pay off in the long run – both for insurance companies as well as policy holders (Michel-Kerjan and Kunreuther, 2011). On the other hand, multi-year contracts will be more expensive as they need to consider future uncertainties (Maynard and Ranger, 2012) and thus can be expected to decrease insurance demand. In addition, one-year contracts give the insurers the flexibility to adapt their business to changing conditions such as changes in flood risk due to climate change.

## 3    Analytical framework and research approach

To understand insurers' engagement or lack of engagement in flood risk reduction, it is important to consider three aspects. First, what types of flood risk reduction measures exist and how they contribute to flood risk reduction. Second, what incentives insurers have to promote those risk reduction measures and third, what are the framing conditions (i.e. insurance scheme, market penetration) which hinder or stimulate insurers to get engaged in flood risk reduction. For the analysis, conceptualization of the flood risk management cycle as used by Surminski and Thieken (2017a) was adapted as it is shown in figure 2 (lower right circle). Changes to the original framework are marked in cursive letters. Under the bullet *prevention* "building codes" were added as another strategy to enforce resilience on a larger spatial scale. The term *mitigation* was replaced by *adaptation* to be consistent in the working of the paper. The term "awareness campaigns" was removed from the category "preparedness for response" and added to the risk assessment phase in the category risk knowledge, as I consider increasing

citizens risk knowledge as the main purpose of awareness campaigns. Without acquiring knowledge of their flood risk i.e. assessment of their own risk, citizens will not take flood risk reduction measures in the "risk reduction phase".

The flood risk management framework was further expanded by postulating that insurers engagement in flood risk reduction have interdependencies with framing conditions. These conditions include the insurance scheme insurers are operating in and the distribution of responsibilities for flood risk management in society, the market penetration of flood insurance and the flood risk situation in a country. In addition, there are also interdependencies with the types of incentives insurers can use to promote flood risk reduction.

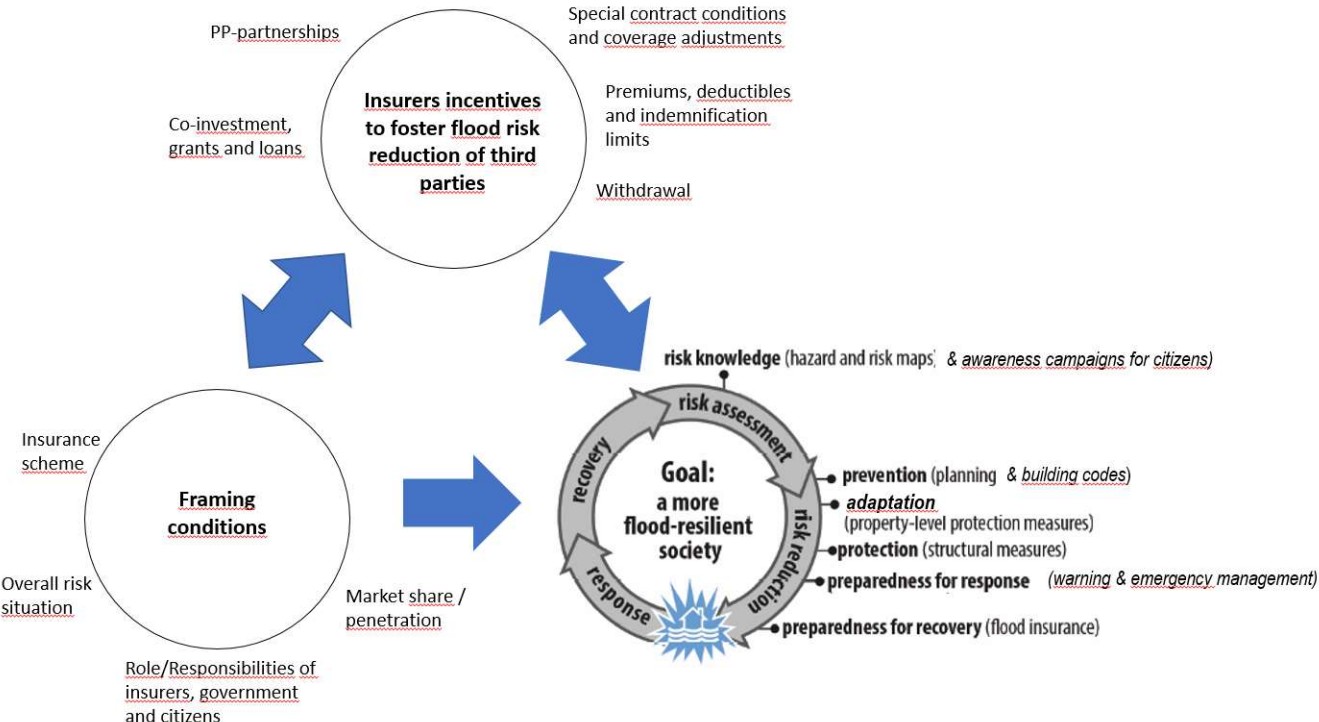

**Figure 2 Assessment framework for interdependencies between insurance schemes, flood risk reduction measures and insurers incentives for flood risk reduction; the flood risk management cycle (lower right corner) was adapted from Surminski and Thieken (2017a)**

To analyse what mechanisms insurers use to incentivize flood risk reduction measures and to evaluate the influence of framing conditions, relevant peer-reviewed scientific literature was examined, found in the research database Web of Science for the combination of search terms "insurance", "flood" and "risk reduction" (46 hits, oldest from 2010) and "insurance", "flood" and "prevention" (38 hits, oldest from 2000), which were present either in the title, keywords or the abstract of the publication. The results of both searches were combined and doublings were removed. After that all abstracts were screened and publications not considered as relevant as they were focusing on developing countries or dealing with different issues such as behavioral studies on risk reduction or loss modelling, were removed. It remained 31 core publications, which were considered

as highly relevant. Additional relevant publications were identified based on the reference lists of these core publications. In addition to academic literature, Google was used to assess what kind of flood risk reduction measures are currently used by insurers in developed countries. A search was made for websites containing information on "insurance" and the different flood reduction measures as listed in the assessment framework in figure 2. The search was performed in English, German and Norwegian language. This work is explorative in nature and the author is aware to the potential bias due to her limitations in language skills.

## 4    Results and discussion

### 4.1  Insurance engagement in flood risk reduction

#### 4.1.1    Risk knowledge

Even though **collecting data** and increasing **knowledge about flood risk** as such does not reduce the flood risk itself, it can be considered as an important precondition for a better understanding of the risk and successful implementation of risk reduction measures. Different forms and various channels of communication are used to **raise awareness** and increase understanding around issues of climate change, flooding, flood risk, adaptation measures and insurance, and how they are all linked (Bouwer et al., 2014). **Communication and education measures targeting citizens** include mass media campaigns in newspapers, radio, TV and internet, but also compulsory information disclosure for rented or sold properties, as well as education programs. They aim to educate citizens about flood risk in general, available market insurance policies, but also to promote decentralized property-level adaptation measures. Insurers have a vital interest in raising public awareness about flood risks, as those campaigns can help to inform the people about their personal flood risk and make them aware of the benefits of insurance and risk reduction (Den et al., 2017). Such campaigns can be considered as part of the interaction between insurers and their actual or future potential policyholders. Thus, they have the potential to increase market penetration to avoid negative selection.

In Germany, information campaigns for natural hazard insurance were run by several federal states with support from the German Insurers Association (GDV). They were quite successful in raising the insurance penetration to on average 40% (GDV, 2017) and managed to double the number of policies within the last 15 years[2]. In France, awareness-raising campaigns are eligible for funding from the Barnier Fund[3], which is financed by insurance premiums. In Switzerland, private insurers use the public hazard maps to combine them with information on how to protect private property from natural hazards[4]. Also in Italy increased insurance activities for offering advice on property-level adaptation measures were noticed (Botzen et al., 2017). Outreach and marketing campaigns are also run by the NFIP in the US (Kousky, 2017). A survey among private insurance

---

[2] http://www.gdv.de/2017/04/mehr-hausbesitzer-versichern-sich-gegen-ueberschwemmungen/
[3] https://www.ccr.fr/en/fonds-publics
[4] https://www.zurich.ch/de/services/naturgefahren/start

companies in the US showed that nearly half of all interviewed property and casualty insurers mailed leaflets or provided information on their website on how to reduce losses from weather related disasters (Leurig and Dlugolecki, 2013). In Norway, in a cooperation of academia and Finance Norway, an umbrella organization for the financial industry including insurers, an education tool for schools[5] was developed to introduce knowledge about climate change, extreme weather events and prevention (Finans Norge, 2012). In the US, information-based activities rank highest for implementation by municipalities among the risk reduction measures, which are awarded by credit points in the NFIP community ranking system (Sadiq and Noonan, 2015).

In a workshop on how to improve the linkages between insurance and flood risk reduction, lack of access to detailed information was mentioned as a barrier (Surminski et al., 2015). Indeed, **information sharing between insurers and governmental actors**, in particular, municipalities were suggested in several studies (Den et al., 2017; NOU 2015:16, 2015). Apparent benefits for both governmental actors, as well as insurers, are that they can improve their knowledge base on flood risk and flood risk reduction by integrating new data sets. For example, in Germany, the insurance industry integrated in cooperation with authorities the official flood hazard zones into their flood zoning system (ZÜRS), which resulted in a more accurate risk classification, and that more households were considered insurable against flooding (Kron, 2013; Surminski and Thieken, 2017a). Another example for information sharing and public-private cooperation with insurance involvement is the HORA online-platform[6] developed in Austria, which provides risk zoning for natural hazards, including floods, and aims to inform citizens (Stiefelmeyer and Hlatky, 2008). Similarly, the Association of British Insurers in the UK cooperated with public agencies to improve the quality of flood maps in the UK (Surminski, 2014). In Switzerland, a large private insurance company combined public natural hazard maps with national economic data and their insurance data to create better risk maps, and made the data publicly accessible[7]. In the US, flood insurance-rate maps developed under the NFIP are regarded as an important risk communication tool (Kousky, 2017). In fact, the whole organization of the insurance system in the US is based on these maps, as the flood zones determine who is required to purchase flood insurance. These maps are evaluated regularly and local governments can get engaged in the mapping process and contribute with better data (Kousky, 2017). In Denmark the combination of insurance and municipal data led to better reinsurance conditions for the insurers as they could proof that municipal efforts have reduced the flood risk (Den et al., 2017). On the other hand, municipalities can also learn more about their exposure by looking at historic insurance claim data, as they did in a pilot project in Norway[8]. Information transfer from insurers to municipalities might be more problematic compared to the other way around. Claim data is spatially explicit, i.e. data privacy issues must be considered. And in private insurance systems, historic claim data has a competitive value and therefore insurers are often reluctant in sharing it (Botzen et al., 2010). In France, where flood insurance is compulsory and provided by private companies backed up with a state guarantee (Porrini and Schwarze, 2014), a part of the insurance revenues

---

[5] https://www.miljolare.no/en/aktiviteter/klima/ekstremver/
[6] http://www.hora.gv.at/
[7] https://www.mobiliar.ch/die-mobiliar/engagement/praevention/mobigis
[8] https://www.finansnorge.no/globalassets/presentasjoner/2017/nordress-island.pdf

from natural hazard insurance are transferred to the Major Natural Risk Prevention Fund, also known as the Barnier Fund[9]. This fund finances actions that decrease the exposure of insurers (Poussin et al., 2013; Suykens et al., 2016). In the context of communication, it finances studies necessary to prepare the obligatory natural disaster prevention plans in municipalities.

An interesting new approach to combine information transfer about property-level risk reduction measures to citizens with data collection on property level was found in Germany. Here the insurance industry was involved in the development of a building certificate ("Hochwasserpass"), which is available since 2014 and proofs the standard of flood-protection for single properties[10]. After a first rough flood risk assessment combined with general information about flood risk and flood risk reduction options, which is available in the internet for free, the property-owner pays for the second step, which includes a more detailed assessment and results in a flood risk certificate. This second step includes plausibility checking of the local conditions and an on-site visit by a qualified expert if necessary (DKKV (ed.), 2015) suggesting how to make the property more flood resistance (Surminski and Thieken, 2017a). The idea behind the certificate is, that it can be used by property-owners in negotiations with insurers for better insurance conditions or with banks for mortgages.

In a similar direction goes a new knowledge sharing initiative, which was launched in 2017 in the UK, where certified surveyors will provide information on building protection measures to property owners, as well as collect data about the effectiveness of these protection measures after an event and enter it into a database shared by several insurance companies[11]. Such a database could in the future deliver the information needed to assess the flood risk of single properties, which is a precondition for better risk-based pricing (see also 4.2.1).

There also exists **communication fora** initiated by insurers or reinsurers meant to provide a discussion platform for different stakeholders to exchange opinions about climate change adaptation, including floods. Examples are the Munich Climate insurance initiative (MCII)[12] in Germany, Climate Wise in the UK[13] or international fora like the Extreme Events and Climate Risk forum hosted by the Geneva Association[14] or the UNEP Finance initiative[15].

### 4.1.2    Prevention

Planning, including building codes  and land-use planning, is considered an important factor in flood risk management as well as in climate change adaptation (Botzen and Van Den Bergh, 2008; Hurlimann and March, 2012; Measham et al., 2011; Petrow et al., 2006). **Land-use planning** is an integral part of flood risk management, as it prevents future detrimental developments that increase flood losses by, for example, avoiding new settlements in flood-prone areas or restricting the use of those areas

---

[9] https://www.ccr.fr/en/fonds-publics
[10] https://www.hochwasser-pass.com/
[11]    https://www.bre.co.uk/news/BRE-Global-to-launch-a-new-certification-scheme-for-property-flood-resilience-surveyors-1217.html

[12] http://www.climate-insurance.org/home/
[13] http://www.cisl.cam.ac.uk/business-action/sustainable-finance/climatewise
[14] https://www.genevaassociation.org/research/topics/climate-risk/
[15] http://www.unepfi.org/

(Botzen and Van Den Bergh, 2008) or by preserving wetlands, which can contribute to reduced flood losses (Brody et al., 2015; Calil et al., 2015). Land-use planning takes place at national, regional and local levels, and across levels (Bouwer et al., 2014; Petrow et al., 2006). Another strategy is the enforcement of **building codes**, which can reduce the vulnerability of buildings to flooding and thus also, damages. There are building codes that are valid for all properties, but in flood-prone areas, building codes are often more strict (Aerts and Botzen, 2011). One important aspect to be considered is, that building codes only apply to new buildings i.e. not to the existing building stock.

The decree and enforcement of building codes and land-use planning procedures are governmental tasks. Due to their importance for flood loss reduction it is not surprising that in countries where the state provides flood insurance, insurers exert influence on land-use planning and formulation of building codes. In the US, the NFIP i.e. the insurer itself, sets minimum requirements for participating municipalities concerning their flood zoning and their buildings codes. They require elevation of all new buildings or when heavily damaged buildings are reconstructed, to build above water levels of the 100-year flood line (Aerts and Botzen, 2011; Kousky, 2017; Petrow et al., 2006). Communities can voluntarily legislate stricter building codes than required by the NFIP. In New York additional requirements for elevation, and wet and dry flood-proofing of buildings exist, which are distinguished according to whether a building lies in a coastal or an inland risk zone (Aerts and Botzen, 2011). While NFIP was evaluated positively for limiting the vulnerability of new buildings, it was criticized for its poor land-use management, i.e. giving incentives to settle in hazard areas, instead of limiting new developments in flood zones (Aerts and Botzen, 2011; Johnston, 2012; Pompe and Rinehart, 2008). In Switzerland, insurers are also involved in the process of enforcing building codes. In areas with a determined risk they will for example, check the building plans and requirements for hazard-adapted building construction (Camenzind and Loat, 2014). Swiss insurers can also significantly influence land-use planning processes: denying coverage in certain areas will lead to an adjustment of land-use plans (Petrow et al., 2006). Similarly, in the UK properties built after 2009 in areas of high flood risk can be excluded from insurance. But even though meant to create a disincentive for further valuable developments in high risk areas, there are still new houses built in floodplains (Surminski and Thieken, 2017b). In Germany´s private insurance system insurers get to a limited extend engaged in the formulation of building codes at a higher-level. The GDV as representative for its members comments e.g. on new building codes or changes in building norms (GDV, private communication).

### 4.1.3    Adaptation by property-level measures

While large-scale structural protection measures aim to reduce the probability of flood occurrence or its strength (i.e. the hazard), **decentralized property-level risk reduction or protection measures** can also reduce the negative consequences of flooding (i.e. exposure and vulnerability). There are a broad variety of structural and non-structural measures: structural single-property building measures as elevation or dry- and wet-proofing of houses, or non-structural strategies, such as removing all valuable items from the basement of a property. Several studies demonstrated the effectiveness of these kinds of adaptation measures especially for high-probability flood events (Bubeck et al., 2012; Hudson et al., 2014; Kreibich et al., 2005, 2011; de Moel et al., 2014; Poussin et al., 2015), but also showed that the effectiveness of each protection measure very much

depended on the type of measures chosen and local circumstances, such as the characteristics of the flood (Hudson et al., 2014; Poussin et al., 2015). It is easy to imagine that mobile water barriers on doors and windows will have no effect once they are surpassed, whereas securing oil tanks against buoyancy will be effective, independent of flood water depth (Kreibich et al., 2011). Structural property-level protection measures can be addressed in building codes, while this is not the case for non-structural measures.

Insurers have different options to get engaged in adaptation activities. In the US, NFIP can require flood-proof construction in new building areas or improved reconstruction of destroyed properties. Within the one-hundred year flood zone buildings have to be elevated above the water depth expected for a one-hundred year flood (Aerts and Botzen, 2011; Kousky, 2017; Petrow et al., 2006). In Germany, with a private insurance system, there is no statutory obligation to install property-level protection measures (Surminski and Thieken, 2017a), but surveys with insurance companies revealed that more and more insurers require property-level protection measures in high-risk zones as a condition for offering insurance (DKKV (ed.), 2015). And even though German insurers in general only pay for "reconstruction to the same conditions as before", more and more insurance companies also allow for flood adapted reconstruction or reinforcement of buildings after severe destruction, when this does not cause higher costs than a standard reconstruction following the applicable building codes (GDV, 2014). A similar rule applies in Germany for relocation, i.e. insurers refund only costs for reconstruction at the same place, additional costs due to relocation are not covered, but if covered from another source they allow for relocation to a safer area (DKKV (ed.), 2015). In Denmark and Iceland insurers can require the implementation of property-level protection measures or otherwise restrict coverage (Den et al., 2017; Priest et al., 2016). Insurers can also positively acknowledge and thus foster implementation of property level risk reduction by designing their products in a way, that risk-reducing behavior is rewarded in form of reduced premiums or deductibles (for more details on this issue see chapter 4.2.1). However, a main barrier for policyholders to implement property-level protection measures is certainly the large upfront-investment required (Aerts and Botzen, 2011; Bräuninger et al., 2011; Kunreuther, 1996). Such an investment must be done all at once, while benefits in the form of lowered premiums or deductibles will accrue over a long period of time. A suggestion to overcome this dilemma is that insurers cooperate with banks or the state provides inexpensive loans or grants to cover the large up-front investment (Botzen and Van Den Bergh, 2008; Kunreuther, 2006). When the monthly repayment for the loan is less than the difference between the insurance premium with and without property-level protection measures, there is a clear financial gain for the policyholder (Bräuninger et al., 2011; Kunreuther, 2015). In practice, only the NFIP in the US offers grants to policyholders to implement property-level protection measures on existing buildings, or to upgrade them to comply with current building codes in the case of severe damages (Kousky, 2017). In case of severe repetitive damages, the policyholder has to implement protection measures to avoid an increase in premiums of 150% (Aerts and Botzen, 2011). Unfortunately, the list of property-level protection measures that are eligible for grants is currently very short. The measures only include elevation of houses, flood-proofing and relocation (Aerts and Botzen, 2011; Kousky, 2017).

### 4.1.4    Protection by large-scale infrastructure

**Large-scale structural flood infrastructure** such as dams, dikes, embankments, reservoirs and polders (controlled retention basins), or slope stabilization measures decrease the risk of flood occurrence or the magnitude of the event and have proven their effectiveness in flooding events (Thieken et al., 2016). Common to all large-scale structural measures is that safety is only provided when well maintained (i.e. do not fail) and when design levels are not exceeded (Thieken et al., 2016). Two main functionalities of such infrastructure should be distinguished: infrastructure that protects an area from being inundated i.e. keeps the water outside (e.g. dams and dikes), and infrastructure which enables extra room for water storage such as reservoirs and polders. The first type of infrastructure only avoids flooding in a certain area and can have adverse consequences downstream as greater levels of water flow there. In addition leads avoidance of flooding often to a self-reinforcing cycle, of protected areas attracting further economic development (i.e. the number and value of assets increases), which in turn leads to the claim of even stronger flood defense (Filatova, 2014) or in case of infrastructure failure or overtopping to very high losses. The second type of infrastructure reduces water volume and is thus beneficial for all downstream areas. This difference is important to consider when pondering up- and downsides to large-scale structural infrastructure.

Usually it is the government who invests in large-scale structural protection measures. To the author´s knowledge there are no insurers to-date which completely finance large-scale structural measures. In Switzerland, where a dual system of public and private natural hazard insurance exists, the Swiss cantons monopoly insurers invest on average 15% of their premium incomes in prevention (Ungern-Sternberg, 2004). Private insurance companies also co-found large-scale structural measures in municipalities[16]. Nevertheless, public investments in risk reduction are much higher in districts with a district insurance monopoly (Paudel, 2012), because under private insurance schemes there will always be the risk of policyholders shifting to another insurance company.

In the US, the NFIP offers incentives for communities to implement large-scale structural protection measures. Communities who joined the NFIP can voluntarily participate in the community rating system, which offers premium reductions for the implementation of a list of flood protection measures. This list also contains dams and levees as large-scale structural protection measures (Sadiq and Noonan, 2015). In case the levee owner i.e. mostly municipalities, can document that the levee satisfies the national requirements and provides protection from the 1/100 years flood, the obligation for house owners to take out insurance can be lifted.

In the UK, a completely private insurance market has existed since 1960, when insurers and the government agreed that insurance should be provided to all households and small businesses under the requirement that the government invests in large-scale flood protection to enable a minimum standard of flood protection of 1/75 (i.e. are protected against floods with a recurrence interval of 75 years). The agreement has been revised several times over the years, and although the new 'Flood Re' insurance scheme still contains this requirement, there is no mechanism defined to monitor compliance (Surminski and Eldridge, 2015). As it currently exists, the agreement has been heavily criticized: after public investments reducing the flood

---

[16] https://www.mobiliar.ch/die-mobiliar/engagement/uebersichtskarte-engagements#?topic=34

risk beyond the 1/75 years flood requirement, insurance premiums remained mainly the same, i.e. the expected benefit in the form of avoided losses was not transferred to society or policyholders, but resulted in additional profit for insurance companies (Penning-Rowsell, 2015). Evidence on this issue is difficult to provide as flood insurance is offered bundled, with other hazards, making it difficult to disentangle the flood proportion of the premium. To overcome this situation, Penning-Rowsell (2015) suggests that insurance companies should be forced to reduce premiums in case that significant risk reductions occurred.

### 4.1.5    Preparedness for response

**Monitoring and early warning activities** encompass meteorological and hydrological observations, often in combination with forecasting models, which allow predictions of approaching hazards. In addition, successful early warning systems require that citizens receive the warning, understand it and react by taking the appropriate **emergency response measures**. Emergency responses measures involve removing mobile items of value from the area at risk (such as cars), that temporary small-scale adaptation measures such as floodgates or sandbags are put in place and that people in the area at risk from flooding are evacuated.

Even though monitoring, early warning and emergency response right after or during a flood are governmental tasks, some insurance companies provide a warning-service for policyholders. For example, the Swiss cantons monopoly insurers offer a mobile phone application that warns the user from approaching natural hazards and at the same time provides safety information on how to reduce losses[17]. The benefit for insurers is that their policyholders then have the possibility to safeguard movable items (e.g. cars, furniture) and thus reduce losses. Insurers become also active as soon as possible after the damage has occurred. During damage appraisal, their employers can often give valuable advice on how to avoid a further increase in damages by properly starting the recovery process. An example from insurance engagement in emergency response activities comes also from Switzerland: The cantonal monopoly insurers finance the Fire Service and Cantonal Civil Defence Services (Atreya et al., 2015).

### 4.2  Insurers incentives for flood risk reduction

### 4.2.1    Premiums, deductibles and indemnification limits

In academic literature, the most frequently mentioned mechanisms to stimulate policyholders to adopt risk-reducing behavior are premiums and deductibles – in case they are reflecting the actual risk a policyholder is exposed to (Botzen and Van Den Bergh, 2008; Kunreuther, 1996). In a perfect market, with well-informed and rational-acting market participants, insurers would earn enough premiums to cover all losses and policyholders would implement flood risk reduction measures when it is economically reasonable for them. A resulting reduction in flood risk would then mean that the insurance requires less money to cover the losses and thus premiums and deductibles could decrease. Reality however, looks a bit different. Firstly, premiums

---

[17] https://wetteralarm.ch

calculations often do not only follow actuarial principles, but are restricted by legislation (Kousky and Shabman, 2014) with the aim to avoid excluding certain policyholders or to avoid rapid price hikes. In France, where the solidarity principle is very important, all citizens pay the same fixed rate determined by the government, independent from the risk they are exposed to (Suykens et al., 2016). Secondly, flood risk is bundled with other risks so the premium does not reflect a single risk. Bundling flood with other natural hazards (France, Portugal, Switzerland and Iceland), fire (Belgium, Denmark) or building/household insurance (US, Spain) is very common (Lamond and Penning-Rowsell, 2014; Maccaferri et al., 2012). Thirdly, premiums are cross-subsidized, either within a peril, between low and high risks, or between perils. For example, in the US, policyholders in low risk areas are charged a higher premium than the one which would be adequate for their risk, and thus subsidize high-risk areas where a risk-reflective premium is considered to be too high to be affordable by policyholders (Kousky, 2017; Kousky and Shabman, 2014). Fourthly, market competition in private markets is so high that insurers keep the premiums artificially low to attract more customers; this is for example the case in the UK (Priest et al., 2016). And fifthly, policyholders or potential policyholders do not always behave rationally; people tend to underestimate risk probability and their need for insurance (Botzen et al., 2013; Kunreuther, 1996). Their decision to purchase insurance relies on their risk perception, previous experience, previous provision of governmental loss compensation, and other factors (Seifert et al., 2013). There is a large body of literature studying behavioral determinants of insurance purchase (see e.g. Botzen and van den Bergh (2012), Bubeck et al. (2013), Kahneman and Tversky (1979), Kunreuther and Pauly (2004), Slovic (1987)), however it would go beyond the scope of this manuscript to discuss this in detail.

In fact, despite being addressed so often in academic literature, risk-reflecting premiums for residential flood insurance seem to be more the exception than the rule (see e.g. table 1 and the country overviews provided by Atreya et al. (2015) and Den et al. (2017)), but this might change in the future. In Germany, where flood insurance is an optional add-on to building insurance and provided by private companies, individual risk-pricing on single property level was very uncommon (Thieken et al., 2006), recently individual agreements to insure single high-risk properties seem to become more common and reduced deductibles and premiums are used to reward property-level protection measures (DKKV (ed.), 2015). In the UK, the new 'Flood Re' scheme hopes to initiate a smooth transition towards risk-based pricing (Surminski and Eldridge, 2015), although at current, risk decrease due to large-scale structural measures on the municipal level is not considered in premium prices (Penning-Rowsell, 2015). Denmark uses a mixed approach of voluntary-compulsory and public-private insurance, which enables them to adopt risk-based pricing for minor flooding events and pool the risk for larger events. Private insurers offer a voluntary risk-based insurance for river and surface flooding after cloudbursts, while losses from storms and storm floods are financed via a tax which is included in the compulsory fire-insurance (Den et al., 2017). The NFIP in the US calculates full-risk premiums (i.e. unsubsidized premiums) by considering risk-zones as well as the type of property, and certain property characteristics such as number of floors, existence of a basement or elevation, and several premium adjustment factors in addition (Kousky, 2017). Such a system could lead to risk-reflecting premiums, but currently the pricing structure is still too coarse, risk-reflection is disturbed by cross-subsidies, and the risk-zoning maps were criticized to be inaccurate; all factors which inhibit reflection of the real risk (Kousky et al., 2016). However, has the NFIP a mechanism to link property-level

protection measures with premiums: It is an requirement of the NFIP that new or reconstructed buildings within the one-hundred year flood zone are elevated above the water depth expected for a one-hundred year flood (Aerts and Botzen, 2011; Kousky, 2017; Petrow et al., 2006). And only with an elevation certificate, which can be issued by state-licensed surveyors, architects or engineers and have to be paid by the policyholder, policyholders are eligible for premium reductions (Aerts and

Botzen, 2011). Similarly uses the NFIP its community rating system (CRS) to incentivize municipalities to implement flood risk reduction measures, which then results in premium reductions for the city inhabitants. But the overall success of incentivizing municipalities seems to be low: by 2014 only 5% of all NFIP communities participated in the CRS (Kousky, 2017). The NFIP makes also use of negative price signals, indicating a potential risk increase in the future. When municipalities do not fulfill NFIPs requirements for flood management even after notification, they are put on probation and can be suspended

from the program in the worst-case scenario i.e. no flood insurance would be available for their inhabitants. In the suspension phase a surcharge is added to each new or renewed policy, aiming to make the policyholders aware of the shortfalls of the municipality (NFIP, undated) or to exert pressure on the municipal government to fulfill the NFIP criteria. A similar mechanism to exert pressure on local governments via the citizens is used in France, where deductibles increase considerably in the case of repeated losses and when the municipality does not develop risk prevention plans. The aim there is that affected

citizens should also lobby the government to implement large-scale structural protection measures, but this mechanism does not really appear to be successful (Poussin et al., 2013; Suykens et al., 2016).

When it comes to the difference between premiums and deductibles, Bräuninger et al. (2011) argue that risk-reflecting deductibles might be far more effective in promoting risk reduction behavior than premiums, when they are in a similar order of magnitude as the costs for property-level protection measures and thus the rentability of an investment becomes more

obvious to the policyholder. Similarly, Smolka (2006) argues that policyholders have to carry a substantial portion of the loss to make deductibles an effective tool to incentivize risk adaptation. He suggests deductibles should be at least 5% of the insured sum or 10% of every loss. Thieken et al. (2006) found that deductibles for households ranged from €500 to €5000, which would mean an incentive of €50 and €500 expected losses in areas with a very high flood probability of 1/10. But making properties flood-proof by sealing doors and raise light wells was reported by Holub and Fuchs (2008) to cost between €2400

and €8400 (2008 prices). Thus, in this example, only for households with the very high flood risk probability of 1/10 and in case the costs of flood-proofing are low, would the high deductible work as a financial incentive to invest in property-level risk reduction. For floods with lower probability this would not be the case, even though the property-level protection measures might be still cost-effective. A recent study of Den et al. (2017) found that the use of deductibles is widespread in Europe, but that these deductibles are relatively small i.e. not in the range of what most property-level protection measures would cost and

thus the incentive given by the deductible must be regarded as limited. A counter argument for the use of deductibles is, that they are uncertain future costs for policyholders and that they will not notice the cost-effectiveness of property-level protection measures in comparison to deductibles before a flood hits them (Priest et al., 2016), whereas premium reductions are more tangible benefits, i.e. the policyholder will notice them each time they pay their premiums.

Indemnification limits, i.e. a capping of the amount of compensation policyholders can receive, are beside deductibles another possibility of sharing the financial burden between insurers and policyholders (Green and Penning-Rowsell, 2004). Indemnity limits as percentage of the property-value insured are practiced in Austria and Italy (Den et al., 2017), as well as in the US (Lamond and Penning-Rowsell, 2014). Loss limits per event or per year are practiced e.g. in Belgium, Iceland and the

Netherlands  (Priest et al., 2016). Indemnification limits can be considered as even less tangible than deductibles, as they only affect policyholders when it comes to high flood losses. This is mainly the case when a low-probability event, i.e. an extremely strong event, hits, so many property-level protection measures such as mobile walls would anyhow fail to protect the property. So, it can be concluded that the use of indemnification limits can be considered as not appropriate to serve as an incentive to promote flood risk adaptation.

When applying risk-adapted prices, insurers have especially in high-risk areas to find a balance between risk-reflecting premiums and acceptability and affordability of insurance by customers (Smolka, 2006). Unaffordable premiums can in fact lead to an exclusion of properties from insurance, which can be intended (see 4.2.3 on withdrawal) or unintended i.e. when it is e.g. a low-income area. A solution to provide also insurance cover for reasonable conditions in low-income areas with an elevated flood risk is to provide inexpensive loans or grant for property-level protection measures, which then in turn also

decrease the insurance premiums (see 4.2.1).

Another issue is the availability of detailed risk information, which is a pre-condition for risk-based flood insurance. In Europe the EU floods directive (EC, 2007) requires that flood hazard and risk maps are prepared in all countries (Nones, 2017). Those maps can form a first basis for partial risk-based pricing based on risk-zones, which is e.g. applied in Germany for residential flood insurance (Atreya et al., 2015). For coming to a more-detailed flood risk assessment down to property-level, insurers

have to find ways to overcome the high transactions costs associated with property-level risk assessment by for example shifting these costs to the policyholder (see example from Germany described in 4.1.3).

### 4.2.2    Special contract conditions and coverage adjustments

Another steering tool insurers use to trigger flood risk reduction are special policy conditions in insurance contracts and coverage adjustments i.e. certain assets are excluded from insurance. In Denmark and Iceland, after a flood event insurance

policies can require implementation of property-level protection measures or coverage will be reduced or insurance completely refused (Priest et al., 2016). In the Netherlands restrictions and exceptions for coverage of losses caused by extreme precipitation exists and are defined in the insurance contracts (Botzen et al., 2010). In Italy insurers exclude goods on ground floor below a certain height from coverage (Fiselier and Oosterberg, 2004), restrictions of coverage in basements are also practiced in Denmark (Den et al., 2017). There are unfortunately currently no studies which investigate in detail how often, in

which countries, for which types of insurance and under which insurance systems insurers make use of the possibility to formulate special policy conditions and adjusted coverage according to risk (Priest et al., 2016).

### 4.2.3    Withdrawal of insurance

Withdrawal of insurance from already existing built up areas should be considered as a last resort solution, even though it might become more likely in the future (Lamond and Penning-Rowsell, 2014). For large areas, this option is neither in the interest of the public, which have to cover potential losses and thus decrease the financial security of households (Botzen et al., 2010), nor an adequate solution for insurers as they must consider this as foregone business (Smolka, 2006). In history, flood insurance withdrawal often led to the creation of national flood loss compensation systems, such as in the US (Thomas and Leichenko, 2011) or in the Netherlands (Suykens et al., 2016). But both examples showed that complete withdrawal from the market is often not a permanent solution: in both countries private flood insurance is available once again – even though it is difficult for private insurers to enter the market again.

For smaller areas or single properties at high-risk, withdrawal must be considered as a reasonable solution to avoid high or repetitive losses. In Australia in 2012 a private insurer temporary withdraw insurance from and two towns, which were flooded 3 times in 2 years causing significant losses. The insurer held a high market share in this region and at that time there were not many competitors offering flood insurance on the market. The 16-month withdrawal resulted the construction of levees by the government (McAneney et al., 2016). The threat of insurance withdrawal is for example used in the NFIP in the US, where policyholders in the one-hundred year flood zone are only eligible for insurance when they and their municipality fulfill certain adaptation obligations (National Flood Insurance Programme, 2012). One the other hand will insurance withdrawal from existing high-risk areas or an announced withdrawal from high-risk areas, which are regulated in land-use planning for high-value uses (e.g. building area or industrial or trade estate), have the long-term societal benefit that high risk areas are kept free or used in a way that no large losses can occur.

### 4.2.4    Co-investment, grants and loans

As discussed before, is the large up-front investment considered a barrier for policyholders, but also smaller municipalities to implement flood risk reduction measures. It is known from the NFIP in the US that policyholders can obtain grants to implement property-level protection measures (Kousky, 2017). In the UK, since 2013/2014, Repair and Renew Grants are available to policyholders, which allow to cover extra costs for a more flood-resilient repair and reconstruction after flood damages (Priest et al., 2016). Michel-Kerjan and Kunreuther (2011) argue that, instead of subsidizing the insurance premiums of low income households living in poorly constructed houses in the one-hundred year flood zone, money should be used to provide them with grants or low-interest loans to implement property-level protection measures, or to even relocate their homes to safer areas. The reduction in insurance premiums could then offset the annual costs of the loan.

The NFIP also provides financial incentives for states and local governments to undertake adaptation activities. Grants are available to support demolition and relocation of buildings or infrastructure, for structural and non-structural retrofitting and elevation of buildings, flood control and prevention projects, as well as for better planning of flood prevention (Aerts and Botzen, 2011). In France, the Barnier Fund also provides financing for municipalities to conduct studies, which are necessary

to develop required risk prevention plans. In addition, the Barnier Fund allows households to apply for subsidies to install property-level protection measures and it provides funds for relocation. According to a survey among French households, which were affected by flooding, the subsidies were hardly used, whereas funds provided for relocation were used more often (Poussin et al., 2013). Co-investment of large-scale structural protection measures by insurers and the government is common in Switzerland and here especially in the cantons with public monopoly insurance.

### 4.2.5 Public-private partnerships (PPPs)

Most existing national flood insurance schemes are based on a PPPs (Paudel, 2012) i.e. public as well as private entities are involved in flood risk management and risk sharing (Atreya et al., 2015; Mysiak and Perez-Blanco, 2016). These arrangements can be very comprehensive i.e. defining the whole insurance scheme in a country as e.g. in the UK but also comprise limited actions in time and space as e.g. the flood insurance promotion campaigns run in cooperation with the federal states in Germany or data sharing agreements between insurers and municipalities in Norway. When it comes to promoting flood risk reduction, Paudel (2012) recommends after reviewing different PPP arrangements that risk reduction should be integrated in the insurance system, while Surminski and Hudson (2017) argue that multi-sectoral partnerships can help to bridge the gap between risk reduction and insurance and that the instrument "insurance" should not be overloaded to fulfil too many different functions.

In the UK, private insurers and the government agreed that private insurers will provide insurance coverage to all dwellings which have a minimum standard of flood protection of 1/75 (Penning-Rowsell et al., 2014). This threshold can be interpreted as an incentive for the government to establish large-scale structural infrastructure to reduce the flood risk below 1/75, and thus enable more households to obtain insurance (Surminski and Eldridge, 2015). It is disputed whether this agreement adequately functions. In the past insurers were criticized for being the main winners of an increase in risk reduction by means of large-scale structural measures whereas the government, i.e. the taxpayers who financed these measures were the main losers (Penning-Rowsell and Pardoe, 2014). In the US, the PPP consists in a governmental flood insurance system, where contacts to policyholders are facilitated by private insurance companies. In Denmark, insurers and the government have several collaborative agreements to address flooding from cloudbursts (Glaas et al., 2016), but they cooperate also in the elaboration of municipal climate adaptation plans (Den et al., 2017). A Swiss insurer cooperated with research partners and NGOs to develop a tool to assess flood risk resilience[18]. PPPs are common in many countries, when it comes to the elaboration of improved flood risk maps based on data shared between insurers and the government (see 4.1.1 for examples).

---

[18] https://www.zurich.com/en/corporate-responsibility/flood-resilience

**Table 2. Overview of insurance engagement in flood risk reduction measures and their use of incentives, contrasted with the framing conditions**

| Type of flood risk reduction measure | Examples of countries where insurers are directly engaged in flood risk reduction or use incentives to promote it | Use of incentives to foster the uptake of flood risk reduction measure by third parties | Influence of framing conditions on insurance engagement in flood risk reduction measures |
|---|---|---|---|
| Risk knowledge (provision) – targeting citizens | US, Germany, Switzerland, Norway | Often done in PPPs, co-financing | Relatively inexpensive, so engagement makes sense under all types of insurance schemes. |
| Risk knowledge (sharing) – with governments | Germany, Norway, Denmark, Switzerland, US, UK, France | Often done in PPPs, co-financing | Easier to share knowledge in countries with public insurance schemes, as claim data has a competitive value in private insurance schemes. |
| Prevention – Land use planning | US, Switzerland | Withdrawal or better the threat of withdrawal is used to enforce better land use planning, e.g. in Switzerland. | No insurance engagement in regional or local land-use planning in private insurance schemes, but in public systems with high market penetration. |
| Prevention – Building codes | US, Switzerland | US: contractual requirement in high-risk zones | Less insurance engagement in private than in public insurance schemes. In countries with private insurance schemes, insurers seem to be only engaged on a higher-level, while in some public systems insurers are also engaged in the enforcement of building codes. |
| Adaptation – property-level measures | Germany, US, Denmark, Iceland | Germany, UK, Denmark, US: adjustment of premiums and deductibles; Denmark, Iceland, Germany: formulation of special contract conditions, coverage restrictions | Requires detailed knowledge of the risk to enable risk-based pricing. Often insurance scheme principles such as affordability or solidarity make the application difficult. Can generate high transaction costs when rolled out to the whole |

| | | | |
|---|---|---|---|
| | | US, France: offering of grants, loans, subsidies | market, so smart solutions are required to evade these costs. |
| Protection – large-scale structural flood infrastructure | Switzerland, UK, US | Switzerland: direct investment<br>US: positive and negative premium adjustments for citizens, when municipality comply with NFIP regulations and e.g. implements infrastructure<br>UK: PPPs, the agreement between the government and the insurers contains the requirement for the government to lower the flood risk to 1/75<br>Australia: insurance withdrawal of from a high-risk area lead to the construction of structural infrastructure by the government | Most insurers currently do not see it as their role to invest in large-scale structural flood infrastructure. In some countries exist mechanisms to stimulate the implementation of large-scale flood protection infrastructure by governmental actors via existing the insurance scheme. |
| Preparedness – monitoring and early warning | Switzerland | Investment to develop early warning applications. | Is considered to be a governmental task, but could be included in risk knowledge provision campaigns for citizens. |
| Preparedness – emergency measures | Switzerland | Co-financing | Is usually considered to be a governmental task. |

## 5    Discussion

Most common across all insurance schemes and countries is the use of information campaigns to raise awareness for flood risk and flood risk reduction among a broader population. These communication measures can be judged to be relatively

inexpensive and easy to perform as most insurers already have the required communication channels in place. They are independent from framing conditions and will get more cost-effective the more people can be reached. A clear downside of communication measures is the difficulty in quantifying their effectiveness e.g. in form of reduced losses. Often a detailed evaluation is completely missing (Kousky, 2017). In addition, continuous campaigns are needed to not revert to the previous status (Den et al., 2017). And even though some campaigns were successful in raising insurance penetration rates as e.g. in

Germany, they did not show an effect on flood risk prevention behavior of policyholders (Osberghaus and Philippi, 2016).

Quite common, although not formalized in most countries, is the exchange or sharing of historic flood risk and loss data between public authorities and insurers with the aim to improve flood risk and hazard maps. While in public insurance schemes the information flow in both directions can be considered as relatively unproblematic, insurers in private insurance schemes might be more restrictive in sharing historic claim data as this data forms part of their business. In this context, trust between

insurers and governmental agencies was found to be an important success factor for sharing data (Den et al., 2017).

Two interesting new approaches were found in Germany and the UK, where the dissemination of information about how to reduce flood risk at property-level to policyholders is combined with the collection of data on property level. The gained data could in the future to enable insurers to better assess the single-property flood risk. The German approach in addition shifts the transaction costs entangled with property-level assessments to the policyholders.

Information campaigns can also be tied to warning services as e.g. done in Switzerland. This makes especially sense for minor flooding events with lower return periods such as e.g. heavy rainfall events. The costs of developing warning applications like mobile phone apps can be considered as low, when it is possible to make use of existing governmental prediction systems. Besides raising awareness engagement in warning activities might also have positive effects for the insurer-policyholder relationship. A main hindrance might also be in this case that early warning and emergency response are in most countries

considered to be governmental tasks.

Limited observations of insurance engagement in preventive activities related to land-use planning and buildings codes might be due to the fact that these activities are in most countries first of all considered to be governmental tasks. Land-use planning in addition occurs mostly at the local to regional level. The transaction costs, i.e. the time and resources it would require for private insurance companies to familiarize themselves with local conditions, might be too high in comparison to the gains

which could be expected by potential loss reductions. In public insurance schemes where insurers operate on cost recovery basis and are not thought to make profits and where the avoidance of overall societal losses is more in the focus of the insurers, engagement in prevention activities will probably be perceived as a useful action to reduce flood losses as the examples from

the US and Switzerland show. Under private insurance schemes it makes more sense that insurance umbrella organizations on behalf of their members follow general developments in land-use planning and get engaged in the development of buildings codes to assure e.g. that the overall loss potential does not increase.

Denial of insurance coverage in high risk areas should be considered as instrument of last resort to influence land-use planning,
even though it can be considered as very effective one. In private markets, the denial of coverage by one company always bears the risk to lose customers to another company. In public insurance schemes, it will depend on the degree of governmental involvement in insurance and probably also the legal regulations if insurers would be allowed to take such a drastic step. Considering the long-term overall societal benefit, it would be better to reduce the accumulation of assets and values in high-risk areas, withdrawal of insurance from certain areas might accelerate this process.

Largely, the main hindrance for increased insurer engagement in large-scale structural measures is probably that insurers currently do not see it as their role to directly provide or invest in risk reduction infrastructure (Surminski et al., 2015; Swiss Re, 2013). From a purely economic point of view, investments in large-scale structural measures are rentable, when the amount of avoided losses exceeds the investment. A precondition for making such kinds of investment profitable for insurance companies is that the policyholders protected by the built infrastructure stay with the same insurer over a long period, i.e. at
least until the investment is paid-off and that a large number of policyholders are protected by the same infrastructure. These preconditions are today only fulfilled in countries with public monopoly insurers and/or where insurance is compulsory, such as Switzerland, Spain or France or quasi-compulsory, as in Ireland and Sweden (Maccaferri et al., 2012).

High transaction costs involved in risk assessment and consideration of adaptation measures at property-level are probably a hinder for insurers to more proactively incentivize property-level protection measures. A solution to overcome this problem is
to shift these costs to the policyholder, by requiring them to pay for experts to assess the flood risk at their property and attest the implementation of property-level protection measures. For insurers organized as mutual insurance the increased transaction costs might be of less importance in case the promotion of property-level adaptation is considered as beneficial for the insurance community.

In private insurance schemes, insurance companies may consider it a disadvantage in the market to require the implementation
of property-level protection from their policyholders, when implementation is not required by their market competitors. This barrier does not exist in public insurance schemes without market competition and might be less pronounced for mutual insurance. In addition are property-level protection measures meant to reduce minor losses from high-frequent events, while insurance should cover large losses from low-probability events (see figure 1), thus most insurers would probably be interested only in incentivizing those types of property-level measures, which are able to reduce also larger losses occurring at more
seldom events such as e.g. safeguarding of tanks to avoid contamination with oil or other hazardous substances, which have shown to increase losses substantially (Kreibich et al., 2005). A good chance to implement property-level protection measures is after severe damages, when reconstruction is required. Here insurance companies could easily not only allow for, but encourage their policyholders to undertake flood adapted reconstruction.

Adaptation by property-level measures is also closely linked to insurance incentives, which can be used to promote them. Even though modelling studies found the use of risk-based premiums to be effective in fostering property-level adaptation measures (Hudson et al., 2016), in practice risk-based pricing is currently more the exception than the rule. First it would require very detailed risk data and information about the effectiveness of different property-level protection measures, which is not

everywhere available yet. Secondly, it would require to disentangle the flood risk and flood risk insurance products from other risks (i.e. other hazards) and products (i.e. sold together with property or fire insurance) it is bundled with and to develop single-risk products. Currently not all countries already provide single-hazard flood risk insurance products (see e.g. table 1). And thirdly all other "premium-corrections" like cross-subsidizing premium earnings in high-risk areas with premium earnings from low-risk areas (as practiced e.g. in the US) need to be removed. This would at the same time mean that the advantages of

bundling or cross-subsidizing such as spreading of risk and increasing affordability get lost. An alternative, which would probably require less re-structuring of insurance products, would be to only give "price signals" to policyholders without calculating risk-reflecting prices i.e. by lowering the premium or deductibles by a certain amount in case property-level protection measures are installed.

In the US and France insurers try to use price signals to policyholders to in parallel incentivize local governments i.e.

municipalities to improve their flood risk management. The idea of engaging municipalities is in principle good as municipalities are often responsible for land-use planning, implementation of large-scale protective infrastructure and emergency response. They have a considerable potential to reduce the local risk of flooding. In both countries, the mechanism works with positive, but also negative incentives i.e. policyholders in the relevant municipality are "punished" with higher premiums or deductibles in case the municipality does not follow up the requirements of the insurer. As a second step, the

mechanism implies that the policyholders in their role of inhabitants of a municipality complain against the municipal flood risk management. As one single complaint would probably not be enough to get things going at municipal level, it would further require that several policyholders join forces. In the opinion of the author there are too many factors that must to come together for the mechanism to work as designed. There are some first indications from France that the mechanism it is not working optimally there (Poussin et al., 2013; Suykens et al., 2016). A detailed evaluation for both France and the US would

be of great value for the future of these programs.

An underestimated instrument to incentivize property-level adaptation is the formulation of special contract conditions. Depending on their market share, insurers are in the unique position of having personalized contact to many property owners, which might positively influence the property owners´ risk reduction behavior. However, from an insurers perspective are individual contract arrangements and on-site risk assessments more rentable for high-value objects such as large business, than

for households or small businesses insurance, which can be considered as mass markets. In private markets with a strong competition, insurance companies might in addition consider obligatory contractual requirements for property-level protection measures as a hinder to increase or hold their market share, as this would require additional efforts from their policyholders. Thus it will be easier to use this instrument in public insurance schemes where competition is lacking (Lamond et al., 2009). For this kind of incentive, it will again probably be beneficial to use "the window of opportunity" after a flooding event to

introduce special contract conditions requiring property-level protection measures. It would be definitely beneficial in society as a whole to include build-back-better requirements as standard element in insurance contracts (Den et al., 2017; Priest et al., 2016).

Another instrument, which requires reconsideration in the authors opinion are deductibles. In Europe, they are currently too small to incentivize the implementation of property-level protection measures considering the price of these measures. But as mentioned before as incentive it might also work to just lower the deductible as a "reward" for implementation of property-level protection measures.

Financial aid for both, property-level protection measures and for large-scale protection infrastructure in form of subsidies and grants was found under public as well as in private insurance schemes. But direct co-investment of insurers in large-scale protection infrastructure only happened in Switzerland. However, for most programs an evaluation of their "success" i.e. the uptake rates or total sums of financial aid used is missing.

In most insurance schemes, public as well as private, there exists already different types of cooperation i.e. PPPs between public and private actors and the public-private relationship spans from "parasitic" to "symbiotic" relations (Green and Penning-Rowsell, 2004). Public insurance in the US, France or Spain make use of the private insurance companies to maintain all forms of customer-insurance relationship. These relationships involving multiple actors are expected to evolve in the future (Surminski and Hudson, 2017) and might also take a role in the promotion of flood risk reduction. National and even local contexts determine if insurance activities are perceived as complimentary or rival to public activities. In a public insurance system where the insurers are closely tied to the government, insurance engagement in land-use planning, warning or investment in large-scale protective infrastructure will be perceived as normal, whereas this probably will not be the case in private insurance schemes. On the other hand, this might be only a question of time as already new forms of financing including private investments are emerging – especially in the area of climate change mitigation – and some of them might be also relevant for flood risk reduction (see e.g. Banhalmi-Zakar et al. (2016)).

## 6    Conclusions

Even though the anchorage of risk reduction is lacking in most insurance schemes (Surminski and Hudson, 2017), this study has revealed that several insurers in developed countries are either directly engaged or use incentives to promote flood risk reduction measures (see table 2 for an overview). These findings indicate that much information about insurance engagement in flood risk reduction is in grey literature or on web-pages in national languages. Thus, there are probably more activities than those reported in this article.

This study advances the existing body of literature by assessing all types of flood risk reduction mechanisms. Direct insurance activities are determined as well as the application of incentives to foster the uptake of flood risk reduction measures by third parties. I also identify which "framing conditions" of insurance activities would be most appropriate. The findings of this study

are relevant for policy-makers when redesigning national insurance systems, for insurers to get inspired by other insurers activities in the field of risk reduction, and for other actors who envisage a cooperation with insurers on risk reduction activities. Surprisingly, risk-based pricing is seldom practiced, even though it is heavily argued for in academic literature. As discussed previously, the key barriers are the lack of detailed information on the single property risk and on the effectiveness of property-level protection measures. The bundling of flood risk insurance products with other products or risks can be considered as another critical barrier. In addition, high transaction costs may prevent more insurance engagement in flood risk reduction. As the knowledge base for detailed flood risk mapping and the effects of property-level protection measures improve or advance in the future, insurers will be able to take them into account in their risk calculations. However, it remains unclear if they would also move to more detailed risk-based pricing and if other mechanisms will be developed to satisfy criteria like the affordability and availability of flood risk insurance. In this context it will be interesting to determine how new solutions like the "outsourcing" of high risks in separate insurance schemes are performing, as done in the UK and Denmark.

There is an indication that the degree of insurance engagement depends on the framing conditions of the national insurance scheme. Insurers in public insurance schemes seem to be more proactive when it comes to flood risk reduction. This is probably due to the fact that the schemes are often interwoven with the government and face less or no market competition. They are also not required to increase their shareholder value. New developments can be expected also in this field: PPPs are already very common in all insurance schemes and it is beyond all doubt that flood risk reduction requires the collaboration of multiple actors from different sectors (Kron, 2015; Surminski and Hudson, 2017). The roles and responsibilities PPPs or multi-sector partnerships could take in different countries in the future will depend on the cultural contexts, historic flood risk management arrangements and the societal roles negotiated for the different actors. There could be a stronger cooperation between banks, insurers and the government to develop new financing solutions. This could include insurance products with a property-level risk reduction component, while taking into account the affordability of those products for all population groups. A cooperation between the building industry and insurers is also possible. Building companies could then already include property-level protection measures in larger buildings projects and offer flood insurance at good terms as a selling point. In the same direction, the findings suggest the creation of more meeting arenas for insurers, regulators, politicians, other governmental actors to learn from each other, build trust and discuss new solutions.

Further detailed investigations are required for assessing the effectiveness of insurers attempts to incentivize governmental actions by changing the insurance conditions for citizens, like in the US or France. In the UK an exclusion from property insurance in high risk areas did not lead to an avoidance of developments. In this case, is it ethically justifiable to penalize third parties (i.e. the policyholders/inhabitants of a municipality) for the shortfall of someone else (i.e. municipal or public administration)? A suggested solution to overcome this is that decision-makers, such as land-use planners, and building companies working in floodplains should retain a share of responsibility and liability for the flood risk (Green and Penning-Rowsell, 2004; Surminski and Thieken, 2017b).

There is a lack of thorough evaluation of insurance activities in order to determine if the activity reached its envisaged effect. This holds true for insurance activities that direct target flood risk reduction measures. Evaluation studies would help cross-

country or cross-insurance system learning and enable insurers to better target their flood risk reduction activities in an effective manner in the future. In this context, a detailed study of the complete Swiss insurance system would be of special interest. Since the Swiss Monopoly insurers are very pro-active when it comes to their engagement in flood risk reduction it would be of great interest to see if there are "spill-over" effects to the private system existing in other cantons. In addition, "real-life" case studies are needed in order to determine the effectiveness of risk-based pricing as tools to incentivize flood risk adaptation at the policyholder level.

Competing interests. The author declares that she has no conflict of interest.

Acknowledgements. This work was funded by the Research Council of Norway, under the grant 235539/E10 (GOVRISK – Governing risk society: Increasing local adaptive capacity by planning and learning networks). I am grateful to the useful comments I got from the GOVRISK- project team, at the NESS conference 2017 in Tampere as well as of the anonymous reviewers and the editor.

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
