# Peer review of "Insurance engagement in flood risk reduction – examples from household and business insurance in developed countries"

_Natural Hazards and Earth System Sciences, 2017_

## Referee Comment (RC1) · Anonymous Referee #1 · 14 Jul 2017

Dear author, thank you very much for your submission. The paper sounds very promising and interesting, but there are several drawbacks, which should be solved within the next version. Especially the research question or aim is quite weak. The author state that the aim was not to create an exhaustive overview of existing initiatives. . ..For a literature review I'm exactly expecting an exhaustive critical overview of the lit. Besides the paper only state "main positive" effects of insurance system, there are some papers by Colin Green and Edmund Penning-Rowsell which see insurance much more critical way; whereabouts insurance are seen as a parasitic system. In particular, the key problems reflects to communities or low-income families which cannot afford insurance bill. I think this needs to add within the paper. Also I'm not entirely sure what is the

aim of chapter 2; I would remove this part and extend on a more critical reflection of insurance system in natural hazards/climate change adaptation. Also a missing point is the method section: please, provide a more detail information how you conduct the survey/review, how you analysed the survey, how you select the used papers etc. On page 13 you talk about successful information sharing: how you define successful, because many people are quite unhappy with the HORA (also often called horrible risk assessment) or Zürs system. Last point what I'm missing is the discussion part of your paper: especially to see if insurance are more efficient in compare to state compensation or if insurance system is more successful in encouragement of the implementation of local adaptation strategies.

---

## Referee Comment (RC2) · Anonymous Referee #2 · 26 Jul 2017

Review on the article "When is it beneficial for insurers to engage in climate change adaptation – a cross country comparison" by Isabel Seifert-Dähnn

The manuscript is enlightening the role of insurers in flood risk management practice and identifies the potential courses of actions of insurers in the field of climate change adaptation. As such, the manuscript is a very interesting work that could address many readers in the field of natural hazards research and risk management practice.

However, the manuscripts does not fulfil the expectations of the reader who is framed by the title and the abstract. The title is misleading. I read the manuscript curiously to get the answer to the question posed in the title. Throughout the document, I could not

find the linkage to climate change adaptation. The manuscript is dealing merely with flood risk reduction and not explicitly with climate change adaptation. In this regard, the manuscript has to be sharpened. As an alternative, the title has to be changed. The conclusions do not state when it is beneficial for insurers to engage in climate change adaptation. However, the manuscript is identifying the field of actions how insurers can lead to contribute to the societal challenges of climate change adaptation. The manuscript is addressing this important gap.

Nevertheless, in this overview of different actions that can be taken by insurers, the manuscript lacks a description of the limitations for insurers. In private markets, insurers have to face regulation by the state (market regulators) or financial regulations (solvency regulations). E.g. price increases, changes in the financial reserves, withdrawals or reductions of coverage may be restricted by market or financial regulators, or – indirectly - by the market itself. Public insurers face even more legislative regulations, and they are closely related to politics. Direct interventions on the ground (e.g. planning and implementation of flood defence measures) are not always an option for insurers because they "rival" (or in the best case complement) the efforts of public authorities and must in any case be allowed/approved/authorized by public authorities. It would be interesting to deepen this framework and analyse the actions that insurers can take and cannot. Other papers also conclude that insurers can only take action in coordination with public authorities (e.g. as cited in the manuscript: Duus-Otterström and Jagers 2011, Keskitalo et al. 2014, Lamond and Penning-Rowswell 2014, Smolka 2006).

The author states that taking influence on individual policy-holders and sustaining them in their adaptation will increase transaction costs of insurers. However, this argument can be seen from a very different point of view: insurers are the only institutions that have a direct cont(r)act with individual customers. This direct contact may be a benefit against other actors and used in climate adaptation, complementary to other actors that address the community as a whole. In my opinion, the manuscript underestimates

the potential of this customer-relationship. This observation may arise from a partial misunderstanding due to a lacking definition of the term "from the insurers perspective".

Furthermore, the author concludes that the roles of insurance and of public authorities must be renegotiated if insurers should become more engaged in adaptation activities. This relates to the title of the manuscript but the author do not provide a proposal for a renegotiation or a direction towards it. Thus, the reader remains unsatisfied. Therefore, the manuscript has clearly to align the title, the aim of the study, the method section, the abstract with the conclusions. I recommend to align the paper along the overview of the different actions that insurers can take and to focus on flood risk reduction instead on climate change adaptation. The latter would require a remarkable extension of the manuscript in regards to the specific literature. An overview of actions is also in line with different statements in the paper (e.g. "This means there is no 'one solution fits all' approach").

The paper also mentions cloud seeding techniques and weather modification. This is a very important point and heavily discussed. However, these statements are not based on provided literature and not re-discussed later in the discussion or conclusion sections. Either the author adds more literature to this point and discuss it in the light of the geoengineering debate and the potential roles of insurers, or this point should not be mentioned in the second chapter.

Another missing point is the role of insurers that are organized in the form of cooperatives. Their business model lays in between of private and public insurers. It should be highlighted that cooperatives are not paying out dividends to shareholders but are able to re-invest their profits in flood defences. The author showed one example but did not draw any conclusion out of it in regard to this potential for climate adaptation.

In sum, the manuscript has to be remarkably re-designed but it is potentially of great interest for the readers of this journal.

In the following, I am enlisting some minor remarks:

[Figure]

P.3, line 10. The subtitle is about flood "protection" whereas the section begins with "flood adaptation"

P.4, ln. 5: provide citation for cloud seeding as flood prevention techniques (or it is for reducing hail losses?)

P. 6, ln. 10: there is some literature about public-private-partnerships...

P.9, ln. 10: one-year contracts may be also seen as a precondition for flexibility and thus allow climate change adaptation and may not act in any case as a barrier for risk prevention. If insurers are bound to very long contracts, they are not allowed to "adapt" their business to future requirements in the strict sense.

P.14, ln.17: This is only a hint: The public insurance company of the Canton of Grisons for example is funding materials for flood interventions of the fire brigades and is financing the elaboration of emergency plans (contingency plans).

P.17, ln.4-7: This a problematic issue and should be discussed more in detail.

---

## Referee Comment (RC3) · Anonymous Referee #3 · 8 Aug 2017

Review of the paper "When is it beneficial for insurers to engage in climate change adaptation - a cross country comparison"

This paper discusses the role that insurers can play in climate change adaptation. It starts with an overview of different flood protection measures, it gives a general introduction to the functioning of insurance markets and then examines insurers' activities in relation to natural hazard risk reduction, which I think is the core of the paper.

In principle the topic is of interest as is clearly argued in the introduction, but I think the title and the abstract do not well reflect the content of the paper. The paper mainly focusses on floods instead of climate change risk in general. Moreover, climate change

adaptation is interpreted as being similar as disaster risk reduction. Most of the paper tends to focus on disaster risk reduction measures and insurance activities in relation to these, and not necessarily on climate change adaptation which implies adaptation to changing risks. This distinction is important because insurance contracts are usually focused on one year, which implies that premiums, deductibles and coverage conditions etc. are determined for the current risk in that year, and not for future risk in a changing climate. So many of the insurance activities in relation to risk reduction discussed in the paper, such as risk based premiums to incentivize adaptation, apply to current risk and may incentivize measures that reduce current risk, but not necessarily apply to adaptation to a changing climate. Also I feel that the promise in the abstract that "it is discussed under which conditions it becomes profitable for them to engage in climate change adaptation" is not really fulfilled in the end, at least not in an in-depth manner.

A general suggestion is to revise the title and abstract so it is better in line with the contents of the paper. Moreover, I suggest to explicitly clarify the research method in the introduction. The literature review used is selective and the paper mainly discusses existing studies of flood insurance systems in a few countries: namely, by mainly focusing on the UK, Germany, France, Norway, and the US. This is in principle not a problem, but this focus and the literature review approach should be clarified upfront. Some of the discussion of these flood insurance systems in these countries is still on a quite general level. It could be considered to provide a more in-depth analysis of a few of these countries to arrive at a more detailed understanding of how insurance contributes to risk reduction there, or not, and how these insurance systems can be improved.

Moreover, I suggest to clarify the innovation of this paper compared to the existing literature on this topic. The main contribution of this paper seems to focus on the relation between flood insurance and risk reduction, but this has already been discussed in several of the studies cited in the paper. Moreover, the main topics addressed and

messages of the paper seem to be very similar to the review paper by Surminski (2014) on the same topic, which is not cited in the paper. In its current form I feel that the paper does not add much new insights to this existing academic literature on the topic of natural disaster insurance and risk reduction. I suggest to clarify in a revision how the paper builds upon existing studies and explicitly indicate what the new lessons are that we learn from this paper.

In addition to these general comments, I list several specific suggestions for improving the paper below.

Page 9: Catastrophe risk models are often used to assess natural hazard risk and determine premiums, instead of only relying on historical loss observations as the text states.

Page 8: Adverse selection results from an information asymmetry; if the individual has better information than the insurer about her/his risk type then the situation may arise that many high risk individuals demand insurance, while insurers do not recognize these high risk types and charge too low premiums to them. It is not trivial in practice that individuals have more knowledge about the natural hazard risk they face, because these are generally low probability events with which individuals have little experience. Moreover, recent advances in catastrophe risk modelling imply that insurers have access to sophisticated risk assessment methods.

Page 8: The French system can also be seen as a public-private natural hazard insurance systems, since there is public reinsurance but private primary insurance.

Page 8: Moral hazard is a term that is often used as a market failure in insurance markets; individuals with insurance coverage may take fewer risk reduction measures if there are information asymmetries and premiums are not risk based. Empirical evidence shows that moral hazard in natural disaster insurance markets is minor; in the contrary the insured tend to prepare better for natural hazard risk than the uninsured (see Hudson et al. 2017). What you call moral hazard is usually defined as charity haz-

ard; due to government compensation of disaster damage people have a lower incentive to insure and take risk mitigation measures (see Raschky and Weck-Hannemann, 2007 for a literature review on charity hazard, and several empirical papers have been published on this topic the last years).

Page 9: "Theoretically, special conditions could be formulated for every individual policy-holder." I think this is a surprising statement. Of course to limit problems with adverse selection and moral hazard theoretical studies have advocated the use of some form of risk based pricing and monitoring of policyholders' risk types, but this does not imply that special conditions have to be formulated for every individual policyholder. Insurers generally work with risk classes for which different premiums and coverage conditions can be specified.

Pages 9-10: I find the discussion in 5.1.1 unclear. A main issue with many large scale structural adaptation measures, like the example of flood protection used in that section, is that they have public good characteristics of being non-rival and non-excludable. It is well known in economics that private markets, including insurance markets, undersupply public goods, which is why their provision is primarily a government task. Of course insurance can stimulate its provision, for example by sharing data and knowledge about high risk areas that need protection or provide lower premiums or better coverage conditions to policyholders in areas where protection measures are installed. However, the section seems to mainly focus on insurance financing and provision of these measures, which is not a logical starting point given our knowledge about public goods.

Section 5.1.2: A main issue with hazard modification measures is the uncertainty of their effectiveness. I don't think these are fully proven or generally accepted methods, which makes it unsurprising that insurers are not involved in this on a large scale. This should be discussed much more critically.

Page 10: I think the main insurance advantage of constructing levees in the US is that

this allows for being mapped out of the 1/100 year flood zone if the levee fulfils that safety standard, which implies that the flood insurance premium declines. This is independent of whether the community participates in the Community Rating System that is described in the text, for which indeed communities can receive premium discounts by engaging in other risk reduction activities than only levees.

Page 10: For the discussion of the UK insurance system it is important to point that insurers regarded flood insurance in high risk areas unviable at current rates if no additional flood protection was installed. Due to the Flood Re agreement indeed insurance is still offered in such high risk areas, but it is not immediately obvious to me why rates should fall if they were initially viewed as being too low given insufficient spending on flood prevention.

Page 11: A few points are relevant to note at the description of the example of Bräuninger et al. (2011). Indeed if the insurance premium is 400 euro, you need a large percentage discount to incentivize risk reduction measures before the discount is sizable in an absolute monetary amount. However, flood-proofing measures are usually only cost-effective in cases when flood probabilities are relatively high (see e.g. Kreibich et al., 2011; Poussin et al., 2015) and in such areas premiums would be a lot higher than 400 euros a year (which would be equivalent to a risk of suffering a 40,000 loss only once in 100 years if premiums are actuarially fair, which they are usually not which implies an even lower flood probability). So in an example where risks are low and flood-proofing measures are not cost-effective, why would one want to incentivize these measures if the objective is to maximize societal welfare? It is only welfare enhancing to do this for policyholders who face a high risk and, hence, pay high premiums in the absence of risk reduction measures, if premiums are partly risk based. An advantage of a premium discount in that case over the mentioned deductible is that the premium discount is a tangible benefit that the policyholder would receive every year. In contrast, the benefit of a lower deductible only occurs when a flood happens which may receive less weight in individual decision making when people underestimate flood

probabilities, as we know is often the case.

Page 12: I don't fully understand why giving a loan for implementing risk mitigation measures would require multi-year insurance contracts. Of course a loan could be combined with a multi-year insurance contract, but couldn't such a loan be equally effective with an annual contract? I think in that case the loan could be equally effective to overcome short term budget constraints of households for investing in risk reduction and spread mitigation costs. The relation of this text with the Bräuninger et al. (2011) example is unclear to me, also given the issues with this example discussed in the comment above.

Page 12: "In private systems, approaches to strengthen private adaptation seem to be in their infancy". I agree this applies to many property insurances for households, but this may be very different for insurances for commercial properties for which it is more common to have on-site inspections to assess risk of insured industries and adjust premiums on the basis of risk management practices. I suggest to be more specific on the basis of which evidence this general claim is made or to make it more nuanced.

Page 13: Another example of the sharing of risk knowledge by insurers may be the Zurich Flood Resilience Alliance in which the insurance company Zurich finances the measurement of flood resilience in communities. See https://www.zurich.com/en/corporate-responsibility/flood-resilience

Page 13: Insurers mainly use pricing on the basis of risk classes to avoid negative selection instead of the awareness campaigns mentioned in section 5.2.2. If awareness is raised then negative selection may still occur if premiums do not sufficiently reflect risk and it is mainly attractive to insure for high risk individuals.

Page 14: "To my knowledge there are no insurers that are currently engaged in emergency response activities." It would be useful to know how exactly this was researched since this general claim seems to contradict the next sentences about insurers giving advice about how to limit damages during/shortly after disasters.

Page 15: "In private insurance systems, I found no evidence for insurers exerting influence on land-use planning or building plan." Also here it would be useful to know on which kind of research this general claim is made. It is surprising since, for example, in the Flood Re system in the UK mentioned before in the paper insurers do not provide coverage to new constructions in high risk flood zones, which aims to steer development away from high risk areas.

Section 5.3.2. I suggest to provide a more in-depth discussion of what role insurers could or should play in liability lawsuits with regards to flood damages. I doubt whether the legal literature on this topic is well reviewed here.

Section 5.5.3. The Flood Re system in the UK is given as an example of insurance agreements with state actors, but not discussed in much detail. As said before in the paper, there are various public-private flood insurance arrangements in different EU countries, which entail different forms of agreements between the state and insurers. I suggest to discuss these in more detail; e.g. what are the key lessons we can learn from these systems for improving insurability and risk reduction?

Section 5.4.1 overlaps to a large degree with the discussion of risk based premiums and deductibles in 5.1.3. I suggest to integrate these texts so the topic is discussed in a more in depth manner. The current text focusses on challenges with incentivizing risk reduction using risk based premiums, which certainty exist. However, the opposite situation may be useful to point out as well; how can one effectively incentivize policyholders to implement expensive risk reduction measures when premiums are not risk based, meaning the policyholder pays the mitigation cost and the insurer receives the benefit in terms of lower expected claim payouts? It is difficult to see the benefits for policyholders to take risk reduction measures in such an insurance system. So even though working with general risk classes for different groups of policyholders and giving premium discounts for some effective measures, like elevation of homes as is done in the US, may not give the perfect signal for risk mitigation, it may be a better alternative than having flat premiums.

Section 5.4.3 overlaps with the earlier discussion about grants for mitigation on page 12. I suggest to integrate these text parts.

References

Hudson, P. et al. (2017). Moral hazard in natural disaster insurance markets: Empirical evidence from Germany and the United States. Land Economics, 93(2): 179-208.

Kreibich, H. et al. (2011). Economic motivation of house-holds to undertake private precautionary measures against floods. Nat. Hazards Earth Syst. Sci. 11: 309–321.

Poussin, J. et al. (2015). Effectiveness of flood damage mitigation measures: Empirical evidence from French flood disasters. Global Environmental Change, 31: 74-84.

Raschky, P. A., Weck-Hannemann, H. (2007). Charity hazard: A real hazard to natural disaster insurance? Environmental Hazards, 7, 321-329.

Surminski, S. (2014). The role of insurance in reducing direct risk – The case of flood insurance. International Review of Environmental and Resource Economics, 7 (3-4): 241-278.

---

## Author Response (AR1)

Dear author, thank you very much for your submission. The paper sounds very promising and interesting, but there are several drawbacks, which should be solved within the next version. Especially the research question or aim is quite weak. The author state that the aim was not to create an exhaustive overview of existing initiatives. . ..For a literature review I'm exactly expecting an exhaustive critical overview of the lit. Besides the paper only state "main positive" effects of insurance system, there are some papers by Colin Green and Edmund Penning-Rowsell which see insurance much more critical way; whereabouts insurance are seen as a parasitic system. In particular, the key problems reflects to communities or low-income families which cannot afford insurance bill. I think this needs to add within the paper. Also I'm not entirely sure what is the aim of chapter 2; I would remove this part and extend on a more critical reflection of insurance system in natural hazards/climate change adaptation. Also a missing point is the method section: please, provide a more detail information how you conduct the survey/review, how you analysed the survey, how you select the used papers etc. On page 13 you talk about successful information sharing: how you define successful, because many people are quite unhappy with the HORA (also often called horrible risk assessment) or Zürs system. Last point what I'm missing is the discussion part of your paper: especially to see if insurance are more efficient in compare to state compensation or if insurance system is more successful in encouragement of the implementation of local adaptation strategies.

Dear Referee 1,

Thank you for your valuable comments. I addressed them in the following way:

- The research aim was reformulated to "This article tries to shed a light on this, by investigating current engagement of insurance in developed countries in different flood risk reduction measures and their use of levers to get other actors engaged. This is discussed against how these activities are influenced by framing conditions such as the insurance scheme or market penetration (see assessment framework depicted in figure 2). The study focuses on developed countries and on household and business flood insurance."

- I did not aim to write a review paper, but collect examples for insurance engagement in risk reduction. But I added a more consistent approach to screen through the scientific literature.

- Affordability of insurance was discussed in more detail, considering also some papers from Green & Penning-Rowsell (even though I must admit that I could not always agree on their view of a "parasitic" insurance system)

- Chapter 2 was removed and relevant parts explaining the functioning of different risk reduction measures were incorporated in chapter 4.
- The method section (3) was revised and figure 2 amended.
- Concerning HORA: the term "successful" was removed, but what you mention here is exactly what I think is currently missing: An evaluation of the existing approaches in different countries to find out what is working or not and what could be done better.
- I separated the results (chapter 4) and discussion (chapter 5) part in two separate chapters and extended the discussion. However, based on the results of this study I do not feel able to make a final judgement on what systems are more efficient to encourage risk reduction.

With best regards, Isabel Seifert-Dähnn

**Answer to Referee 2**

Nat. Hazards Earth Syst. Sci. Discuss.,
https://doi.org/10.5194/nhess-2017-236-RC2, 2017
Review on the article "When is it beneficial for insurers to engage in climate change adaptation – a cross country comparison" by Isabel Seifert-Dähnn

The manuscript is enlightening the role of insurers in flood risk management practice and identifies the potential courses of actions of insurers in the field of climate change adaptation. As such, the manuscript is a very interesting work that could address many readers in the field of natural hazards research and risk management practice.

However, the manuscripts does not fulfil the expectations of the reader who is framed by the title and the abstract. The title is misleading. I read the manuscript curiously to get the answer to the question posed in the title. Throughout the document, I could not find the linkage to climate change adaptation. The manuscript is dealing merely with flood risk reduction and not explicitly with climate change adaptation. In this regard, the manuscript has to be sharpened. As an alternative, the title has to be changed. The conclusions do not state when it is beneficial for insurers to engage in climate change adaptation. However, the manuscript is identifying the field of actions how insurers can lead to contribute to the societal challenges of climate change adaptation. The manuscript is addressing this important gap.

Nevertheless, in this overview of different actions that can be taken by insurers, the manuscript lacks a description of the limitations for insurers. In private markets, insurers have to face regulation by the state (market regulators) or financial regulations (solvency regulations). E.g. price increases, changes in the financial reserves, withdrawals or reductions of coverage may be restricted by market or financial regulators, or – indirectly - by the market itself. Public insurers face even more legislative regulations, and they are closely related to politics. Direct interventions on the ground (e.g. planning and implementation of flood defence measures) are not always an option for insurers because they "rival" (or in the best case complement) the efforts of public authorities and must in any case be allowed/approved/authorized by public authorities. It would be interesting to deepen this framework and analyse the actions that insurers can take and cannot. Other papers also conclude that insurers can only take action in coordination with public authorities (e.g. as cited in the manuscript: Duus-Otterström and Jagers 2011, Keskitalo et al. 2014, Lamond and Penning-Rowswell 2014, Smolka 2006).

The author states that taking influence on individual policy-holders and sustaining them in their adaptation will increase transaction costs of insurers. However, this argument can be seen from a very different point of view: insurers are the only institutions that have a direct cont(r)act with individual customers. This direct contact may be a benefit against other actors and used in climate adaptation, complementary to other actors that address the community as a whole. In my opinion, the manuscript underestimates the potential of this customer-relationship. This observation may arise from a partial misunderstanding due to a lacking definition of the term "from the insurers perspective". Furthermore, the author concludes that the roles of insurance and of public authorities must be renegotiated if insurers should become more engaged in adaptation activities. This relates to the title of the manuscript but the author do not provide a proposal for a renegotiation or a direction towards it. Thus, the reader remains unsatisfied. Therefore, the manuscript has clearly to align the title, the aim of the study, the method section, the abstract with the conclusions. I recommend to align the paper along the overview of the different actions that insurers can take and to focus on flood risk reduction instead on climate change adaptation. The latter would require a remarkable extension of the manuscript in regards to the specific literature. An overview of actions is also in line with different statements in the paper (e.g. "This means there is no 'one solution fits all' approach").

The paper also mentions cloud seeding techniques and weather modification. This is a very important point and heavily discussed. However, these statements are not based on provided literature and not re-discussed later in the discussion or conclusion sections. Either the author adds more literature to this point and discuss it in the light of the geoengineering debate and the potential roles of insurers, or this point should not be mentioned in the second chapter.

Another missing point is the role of insurers that are organized in the form of cooperatives. Their business model lays in between of private and public insurers. It should be highlighted that cooperatives are not paying out dividends to shareholders but are able to re-invest their profits in flood defences. The author showed one example but did not draw any conclusion out of it in regard to this potential for climate adaptation.

In sum, the manuscript has to be remarkably re-designed but it is potentially of great interest for the readers of this journal.

In the following, I am enlisting some minor remarks:

P.3, line 10. The subtitle is about flood "protection" whereas the section begins with "flood adaptation"

→ *This was corrected.*

P.4, ln. 5: provide citation for cloud seeding as flood prevention techniques (or it is for reducing hail losses?) → *The cloud seeding was completely removed.*

P. 6, ln. 10: there is some literature about public-private-partnerships. . .

→ *After restructuring the paper there is an own section on PPPs (4.2.5)*

P.9, ln. 10: one-year contracts may be also seen as a precondition for flexibility and thus allow climate change adaptation and may not act in any case as a barrier for risk prevention. If insurers are bound to very long contracts, they are not allowed to "adapt" their business to future requirements in the strict sense.

→ *I added this argument to the text (chapter 2)*

P.14, ln.17: This is only a hint: The public insurance company of the Canton of Grisons for example is funding materials for flood interventions of the fire brigades and is financing the elaboration of emergency plans (contingency plans).

→ *I added that example, thanks!*

P.17, ln.4-7: This a problematic issue and should be discussed more in detail.

→ *I discussed that a bit more in detail in the discussion-chapter (5).*

Dear Referee 2,

Thank you for your valuable comments. I addressed them in the following way:

- The title of the article was changed and the whole manuscript was focused on flood risk reduction.
- Limitations of insurers: Beyond the type of insurance scheme I did not feel competent enough to discuss further how legislative requirements (market and financial) support or hinder insurers engagement in flood risk reduction, as law is not my field of competence. I mention this limitation also in the manuscript.
- Rival/complementing activities of insurers and public authorities: This is an interesting aspect! I took that up in the discussions-chapter (5).
- Customer-insurance relationship: I discussed that also in more detail in chapter 5.
- Renegiotiation of roles in flood risk reduction: I changed that text and explained in the frame of PPP how different public and private actors including the insurers could work together.
- The paragraphs on cloud-seeding techniques were taken out, as they were not considered to be relevant anymore when focusing the manuscript on flood risk reduction.
- Insurance as cooperatives: I shortly addressed this issue in chapter 5
- Minor remarks: Please see my remarks *after the arrow* →

P.3, line 10. The subtitle is about flood "protection" whereas the section begins with "flood adaptation"
→ *This was corrected.*
P.4, ln. 5: provide citation for cloud seeding as flood prevention techniques (or it is for reducing hail losses?) → *The cloud seeding was completely removed.*
P. 6, ln. 10: there is some literature about public-private-partnerships. . .
→ *After restructuring the paper there is an own section on PPPs (4.2.5)*
P.9, ln. 10: one-year contracts may be also seen as a precondition for flexibility and thus allow climate change adaptation and may not act in any case as a barrier for risk prevention. If insurers are bound to very long contracts, they are not allowed to "adapt" their business to future requirements in the strict sense.
→ *I added this argument to the text (chapter 2)*
P.14, ln.17: This is only a hint: The public insurance company of the Canton of Grisons for example is funding materials for flood interventions of the fire brigades and is financing the elaboration of emergency plans (contingency plans).
→ *I added that example, thanks!*
P.17, ln.4-7: This a problematic issue and should be discussed more in detail.
→ *I discussed that a bit more in detail in the discussion-chapter (5).*

With best regards, Isabel Seifert-Dähnn

**Answer to Referee 3**

Nat. Hazards Earth Syst. Sci. Discuss.,
https://doi.org/10.5194/nhess-2017-236-RC3, 2017
Review of the paper "When is it beneficial for insurers to engage in climate change adaptation - a cross country comparison"

Dear Referee 3,

Thank you for your valuable comments. Please find below in *cursive letters after the arrow (→)* how I addressed the changes you suggested.

With best regards from Oslo, Isabel

This paper discusses the role that insurers can play in climate change adaptation. It starts with an overview of different flood protection measures, it gives a general introduction to the functioning of insurance markets and then examines insurers' activities in relation to natural hazard risk reduction, which I think is the core of the paper. In principle the topic is of interest as is clearly argued in the introduction, but I think the title and the abstract do not well reflect the content of the paper.
→ *The complete article was revised and has now a focus on flood risk reduction. According to that the title was changed and the abstract revised.*

The paper mainly focusses on floods instead of climate change risk in general. Moreover, climate change adaptation is interpreted as being similar as disaster risk reduction. Most of the paper tends to focus on disaster risk reduction measures and insurance activities in relation to these, and not necessarily on climate change adaptation which implies adaptation to changing risks. This distinction is important because insurance contracts are usually focused on one year, which implies that premiums, deductibles and coverage conditions etc. are determined for the current risk in that year, and not for future risk in a changing climate. So many of the insurance activities in relation to risk reduction discussed in the paper, such as risk based premiums to incentivize adaptation, apply to current risk and may incentivize measures that reduce current risk, but not necessarily apply to adaptation to a changing climate. Also I feel that the promise in the abstract that "it is discussed under which conditions it becomes profitable for them to engage in climate change adaptation" is not really fulfilled in the end, at least not in an in-depth manner. → *I tried to improve this and separated the results and discussion chapter. The new manuscript has a more narrow focus on flood risk reduction.*

A general suggestion is to revise the title and abstract so it is better in line with the contents of the paper. Moreover, I suggest to explicitly clarify the research method in the introduction. The literature review used is selective and the paper mainly discusses existing studies of flood insurance systems in a few countries: namely, by mainly focusing on the UK, Germany, France, Norway, and the US. This is in principle not a problem, but this focus and the literature review approach should be clarified upfront. Some of the discussion of these flood insurance systems in these countries is still on a quite general level. It could be considered to provide a more in-depth analysis of a few of these countries to arrive at a more detailed understanding of how insurance contributes to risk reduction there, or not, and how these insurance systems can be improved. → *Chapter 3 contains more detailed information on the research method used. I performed again a more systematic literature search, but I decided not to focus on selected countries as suggested as my aim was to identify as many different practices of flood risk reduction or levers to promote them as possible. What I did according to a suggestion from another referee is, that I narrowed my search down to developed countries and household and business insurance.*

Moreover, I suggest to clarify the innovation of this paper compared to the existing literature on this topic. The main contribution of this paper seems to focus on the relation between flood insurance and risk reduction, but this has already been discussed in several of the studies cited in the paper. Moreover, the main topics addressed and messages of the paper seem to be very similar to the review paper by Surminski (2014) on the same topic, which is not cited in the paper. In its current form I feel that the paper does not add much new insights to this existing academic literature on the topic of natural disaster insurance and risk reduction. I suggest to clarify in a revision how the paper builds upon existing studies and explicitly indicate what the new lessons are that

we learn from this paper. → *Sorry, I was not aware of the Surminski (2014) paper, before I got your comments and indeed some of the issues we take up are similar. The main difference is that I was interested how the type of flood risk reduction, levers to promote third party implementation and framing conditions like the insurance scheme interact. I also performed another literature search (see method chapter 3) to be sure that I did not overlook other important papers.*

In addition to these general comments, I list several specific suggestions for improving the paper below.

Page 9: Catastrophe risk models are often used to assess natural hazard risk and determine premiums, instead of only relying on historical loss observations as the text states. → *I corrected this.*

Page 8: Adverse selection results from an information asymmetry; if the individual has better information than the insurer about her/his risk type then the situation may arise that many high risk individuals demand insurance, while insurers do not recognize these high risk types and charge too low premiums to them. It is not trivial in practice that individuals have more knowledge about the natural hazard risk they face, because these are generally low probability events with which individuals have little experience. Moreover, recent advances in catastrophe risk modelling imply that insurers have access to sophisticated risk assessment methods. → *I improved the definition of adverse selection.*

Page 8: The French system can also be seen as a public-private natural hazard insurance systems, since there is public reinsurance but private primary insurance. → *I corrected this.*

Page 8: Moral hazard is a term that is often used as a market failure in insurance markets; individuals with insurance coverage may take fewer risk reduction measures if there are information asymmetries and premiums are not risk based. Empirical evidence shows that moral hazard in natural disaster insurance markets is minor; in the contrary the insured tend to prepare better for natural hazard risk than the uninsured (see Hudson et al. 2017). What you call moral hazard is usually defined as charity hazard; due to government compensation of disaster damage people have a lower incentive to insure and take risk mitigation measures (see Raschky and Weck-Hannemann, 2007 for a literature review on charity hazard, and several empirical papers have been published on this topic the last years). → *I added the charity hazard to this chapter (chapter 2 in the new manuscript draft) and provided definitions for both terms.*

Page 9: "Theoretically, special conditions could be formulated for every individual policy-holder." I think this is a surprising statement. Of course to limit problems with adverse selection and moral hazard theoretical studies have advocated the use of some form of risk based pricing and monitoring of policyholders' risk types, but this does not imply that special conditions have to be formulated for every individual policyholder. Insurers generally work with risk classes for which different premiums and coverage conditions can be specified. → *What I wanted to point out here was that insurers in principal have the option to formulate individual contract conditions, but that this is often not feasible in practice as transaction costs would get too high. I changed the formulation a bit, so hope this became clearer.*

Pages 9-10: I find the discussion in 5.1.1 unclear. A main issue with many large scale structural adaptation measures, like the example of flood protection used in that section, is that they have public good characteristics of being non-rival and non-excludable. It is well known in economics that private markets, including insurance markets, undersupply public goods, which is why their provision is primarily a government task. Of course insurance can stimulate its provision, for example by sharing data and knowledge about high risk areas that need protection or provide lower premiums or better coverage conditions to policyholders in areas where protection measures are installed. However, the section seems to mainly focus on insurance financing and provision of these measures, which is not a logical starting point given our knowledge about public goods. → *I did not completely get your critiques here. In my opinion the characteristic of "non-excludable" is not always met as protective infrastructure always requires a decision what area is protected i.e. included and what area*

*is not protected i.e. excluded, but even though large scale flood protection measures is considered as public good, why could/should e.g. public insurers not finance them? Please let me know if I misunderstood something here!*

Section 5.1.2: A main issue with hazard modification measures is the uncertainty of their effectiveness. I don't think these are fully proven or generally accepted methods, which makes it unsurprising that insurers are not involved in this on a large scale. This should be discussed much more critically. → *I took that out when I focused the paper more on flood risk. But I agree with you that influencing the weather is a highly uncertain activity.*

Page 10: I think the main insurance advantage of constructing levees in the US is that this allows for being mapped out of the 1/100 year flood zone if the levee fulfils that safety standard, which implies that the flood insurance premium declines. This is independent of whether the community participates in the Community Rating System that is described in the text, for which indeed communities can receive premium discounts by engaging in other risk reduction activities than only levees. → *I made some research on that again and not only the premiums decline, but the obligation to insure against flooding is lifted when protection against the 1/100 year flood is proven.*

Page 10: For the discussion of the UK insurance system it is important to point that insurers regarded flood insurance in high risk areas unviable at current rates if no additional flood protection was installed. Due to the Flood Re agreement indeed insurance is still offered in such high risk areas, but it is not immediately obvious to me why rates should fall if they were initially viewed as being too low given insufficient spending on flood prevention. → *I think the critique was that even though with the use of additional flood protection measures the risk was reduce below the required protection level of 1/75, no changes in premiums were observed. I tried to make this more clear in the text.*

Page 11: A few points are relevant to note at the description of the example of Bräuninger et al. (2011). Indeed if the insurance premium is 400 euro, you need a large percentage discount to incentivize risk reduction measures before the discount is sizable in an absolute monetary amount. However, flood-proofing measures are usually only cost-effective in cases when flood probabilities are relatively high (see e.g. Kreibich et al., 2011; Poussin et al., 2015) and in such areas premiums would be a lot higher than 400 euros a year (which would be equivalent to a risk of suffering a 40,000 loss only once in 100 years if premiums are actuarially fair, which they are usually not which implies an even lower flood probability). So in an example where risks are low and flood-proofing measures are not cost-effective, why would one want to incentivize these measures if the objective is to maximize societal welfare? It is only welfare enhancing to do this for policyholders who face a high risk and, hence, pay high premiums in the absence of risk reduction measures, if premiums are partly risk based. An advantage of a premium discount in that case over the mentioned deductible is that the premium discount is a tangible benefit that the policyholder would receive every year. In contrast, the benefit of a lower deductible only occurs when a flood happens which may receive less weight in individual decision making when people underestimate flood probabilities, as we know is often the case. → *I deleted the premium -example from Bräuninger et al. (2011), unfortunately I found no other numbers on insurance premiums, but I added an example for deductibles. Concerning cost-effectiveness of measures: In my opinion this depends on the measure for which flood probability they will be cost-effective. There are some measures as safeguarding tanks to avoid contamination, which are also effective during strong and seldom events (I discuss that in the new chapter 5). I also added your point with the premium-reductions being more tangible benefits than deductibles (section 4.2.1).*

Page 12: I don't fully understand why giving a loan for implementing risk mitigation measures would require multi-year insurance contracts. Of course a loan could be combined with a multi-year insurance contract, but couldn't such a loan be equally effective with an annual contract? I think in that case the loan could be equally effective to overcome short term budget constraints of households for investing in risk reduction and spread mitigation costs. The relation of this text with the Bräuninger et al. (2011) example is unclear to me, also given the issues with this example discussed in the

comment above. → *When I wrote this my idea was that insurers should provide those loans in combination with the insurance contract. But taking into account also the comments from the other referees I agree that this must not necessarily be the case and thus would also not necessarily require multiple-year contracts.*

Page 12: "In private systems, approaches to strengthen private adaptation seem to be in their infancy". I agree this applies to many property insurances for households, but this may be very different for insurances for commercial properties for which it is more common to have on-site inspections to assess risk of insured industries and adjust premiums on the basis of risk management practices. I suggest to be more specific on the basis of which evidence this general claim is made or to make it more nuanced. → *I deleted this statement.*

Page 13: Another example of the sharing of risk knowledge by insurers may be the Zurich Flood Resilience Alliance in which the insurance company Zurich finances the measurement of flood resilience in communities. See https://www.zurich.com/en/corporate-responsibility/flood-resilience → *I added that example, thanks!*

Page 13: Insurers mainly use pricing on the basis of risk classes to avoid negative selection instead of the awareness campaigns mentioned in section 5.2.2. If awareness is raised then negative selection may still occur if premiums do not sufficiently reflect risk and it is mainly attractive to insure for high risk individuals. → *This is true, but this is only one example. Awareness campaigns can have quite different purposes.*

Page 14: "To my knowledge there are no insurers that are currently engaged in emergency response activities." It would be useful to know how exactly this was researched since this general claim seems to contradict the next sentences about insurers giving advice about how to limit damages during/shortly after disasters. → *I used maybe a too narrow definition of emergency i.e. as activities happening before/during the flood. So I changed the text on this issue.*

Page 15: "In private insurance systems, I found no evidence for insurers exerting influence on land-use planning or building plan." Also here it would be useful to know on which kind of research this general claim is made. It is surprising since, for example, in the Flood Re system in the UK mentioned before in the paper insurers do not provide coverage to new constructions in high risk flood zones, which aims to steer development away from high risk areas. → *I relativized this answer and added also the UK example.*

Section 5.3.2. I suggest to provide a more in-depth discussion of what role insurers could or should play in liability lawsuits with regards to flood damages. I doubt whether the legal literature on this topic is well reviewed here. → *I decided to exclude this topic completely as my legal competence is quite limited.*

Section 5.5.3. The Flood Re system in the UK is given as an example of insurance agreements with state actors, but not discussed in much detail. As said before in the paper, there are various public-private flood insurance arrangements in different EU countries, which entail different forms of agreements between the state and insurers. I suggest to discuss these in more detail; e.g. what are the key lessons we can learn from these systems for improving insurability and risk reduction? → *I merged parts of the content of this section with a new section on Public-Private Partnerships (4.2.5). The PPs arrangements can have quite different purposes, so I do not feel that I can draw a final conclusion on their impact on insurability or risk reduction.*

Section 5.4.1 overlaps to a large degree with the discussion of risk based premiums and deductibles in 5.1.3. I suggest to integrate these texts so the topic is discussed in a more in depth manner. The current text focusses on challenges with incentivizing risk reduction using risk based premiums, which certainty exist. However, the opposite situation may be useful to point out as well; how can one effectively incentivize policyholders to implement expensive risk reduction measures when premiums are not risk based, meaning the policyholder pays the mitigation cost and the insurer receives the benefit in terms of lower expected claim payouts? It is difficult to see the benefits for policyholders to take risk reduction measures in such an insurance system. So even though working with general risk classes for different groups of policyholders and giving

premium discounts for some effective measures, like elevation of homes as is done in the US, may not give the perfect signal for risk mitigation, it may be a better alternative than having flat premiums. → *I tried to remove overlaps between those sections, when restructuring the manuscript. In the new discussion chapter I suggested that in case the premium is not risk-based it might be an option to give "price signals" in form of premium or deductible reductions even though not reflecting the real risk.*

Section 5.4.3 overlaps with the earlier discussion about grants for mitigation on page 12. I suggest to integrate these text parts. → *I tried to remove these doublings.*

With best regards, Isabel Seifert-Dähnn

References
Hudson, P. et al. (2017). Moral hazard in natural disaster insurance markets: Empirical evidence from Germany and the United States. Land Economics, 93(2): 179-208.
Kreibich, H. et al. (2011). Economic motivation of house-holds to undertake private precautionary measures against floods. Nat. Hazards Earth Syst. Sci. 11: 309–321.
Poussin, J. et al. (2015). Effectiveness of flood damage mitigation measures: Empirical evidence from French flood disasters. Global Environmental Change, 31: 74-84.
Raschky, P. A., Weck-Hannemann, H. (2007). Charity hazard: A real hazard to natural disaster insurance? Environmental Hazards, 7, 321-329.
Surminski, S. (2014). The role of insurance in reducing direct risk – The case of flood insurance. International Review of Environmental and Resource Economics, 7 (3-4): 241-278.
**Answer to the Editors comments**

I read the papers you suggested on mountain hazards in Austria and found it interesting to see, that many of the issues I addressed in my paper also apply for other hazards than flood. Nevertheless, I decided not to further consider them in the current version of my manuscript as I found them difficult to fit them into my current line of argumentation.

**Marked up manuscript with relevant changes**

As I completely restructured the manuscript e.g. by integrating the relevant parts of the old chapter 2 into other chapters and giving it a stronger focus on flood risk reduction, it was not possible for me to work with one manuscript-version showing all changes I made. I worked myself successively through all reviewer comments, to assure that I did not forget one, so all major changes are documented in the answers to the referees.

---

## Author Response (AR2)

Review on the article "Insurance engagement in flood risk reduction – examples from household and business insurance in developed countries" by Isabel Seifert-Dähnn

The manuscript now improved a lot. The manuscript now synthesizes the potential measures for flood risk reduction that insurers can take. In comparison to the first version, the title, the research question, the method description and the conclusions are now streamlined. Thus, I do not see any relevant issues that prevent the publication of the manuscript. I have solely a few minor remarks:

- the author is writing in plural (we analysed, our analysis). It might be better to add the co-author or to change this and to write about herself (I analyzed). → I corrected this, sorry, I am not used to write papers alone.

-Table 1: The definition of "all hazards" is vague. In some countries landslides or earthquakes are not included while for example floods, snow avalanches and rockfall processes are included. Please specify the types of processes considered (to be preferred) or write about "multiple hazards" instead of "all hazards" (not preferable). → The table got very crowded when I added the type of hazards and in addition I considered the information on the types of hazard of less relevant for the content of the paper. So I decided to change the wording in the table from "all hazards" to "multiple hazards".

-there are a number of typos → I revised the whole document again, so I hope I found all of them

-page 9, line 30: you are writing how you adapted the figure but not why. → I added some explanations why I adapted the figure

-page 12, line 30: data protection issues --> data privacy issues → I corrected this

-you mention "building codes" in chapter 4.1.2 and "property level measures" in 4.1.3. These terms both are the same matter. In my view, the building code regulates whether object protection is needed or not and thus, both are related. Please specify the division of both in detail or merge them into one chapter. →In my opinion, it is not completely the same, even though there is a large overlap concerning all measures which require changing the structure of the building. But measures like avoiding having high-value non-waterproof assets in flood-prone parts of a house, store chemicals in the upper parts, move valuable items in case of an emerging flood i.e. all non-structural measures are not covered by building codes. In addition, apply building codes only to new buildings and not to the existing stock.

-in page 19, line 30-35 you mentioned the potential of deductibles as incentives. However, the discussion section does not mention this very important point (may somewhere at line 15, page 28). → I took that up in more detail in the discussions-part.

- the conclusion section needs a grammar revision and a thorough check for typos. → I had an English native speaker (American) revising this chapter again

-page 29, line 7: Financial aid for both, property-level protection measures and for large-scale protection infrastructure in form of subsidies and grants was found under public as well as in private insurance schemes. → I changed the wording according to your

suggestion, the complete paragraph reads now like this: Financial aid for both, property-level protection measures and for large-scale protection infrastructure in form of subsidies and grants was found under public as well as in private insurance schemes. But direct co-investment of insurers in large-scale protection infrastructure only happened in Switzerland. However, for most programs an evaluation of their "success" i.e. the uptake rates or total sums of financial aid used is missing.

-maybe the most critical point to consider is that you often mentioned the high amount of transaction costs of implementing flood risk reduction strategies. However, I do not have seen the proof of this statement. Please add references regarding the transaction costs or soften this statement. → I softened this statement a bit as I could not find literature on it related to hazard insurance.